# Threat management priorities for conserving Antarctic biodiversity

**Jasmine R. Lee**[1,2,3,4]*, **Aleks Terauds**[5], **Josie Carwardine**[2], **Justine D. Shaw**[1,5], **Richard A. Fuller**[1], **Hugh P. Possingham**[1,6], **Steven L. Chown**[7], **Peter Convey**[4,8], **Neil Gilbert**[9], **Kevin A. Hughes**[4], **Ewan McIvor**[5], **Sharon A. Robinson**[10,11], **Yan Ropert-Coudert**[12], **Dana M. Bergstrom**[5,8,10], **Elisabeth M. Biersma**[4,13], **Claire Christian**[14], **Don A. Cowan**[15], **Yves Frenot**[16], **Stéphanie Jenouvrier**[17], **Lisa Kelley**[18], **Michael J. Lee**[19], **Heather J. Lynch**[20], **Birgit Njåstad**[21], **Antonio Quesada**[22], **Ricardo M. Roura**[14], **E. Ashley Shaw**[23], **Damon Stanwell-Smith**[18,24], **Megumu Tsujimoto**[25,26], **Diana H. Wall**[27], **Annick Wilmotte**[28], **Iadine Chadès**[2]

1 School of Biological Sciences, The University of Queensland, Brisbane, Queensland, Australia, 2 CSIRO, Dutton Park, Queensland, Australia, 3 School of Biological Sciences, Monash University, Melbourne, Victoria, Australia, 4 British Antarctic Survey, NERC, High Cross, Cambridge, United Kingdom, 5 Australian Antarctic Division, Department of Climate Change, Energy, the Environment and Water, Kingston, Tasmania, Australia, 6 The Nature Conservancy, Arlington, Virginia, United States of America, 7 Securing Antarctica's Environmental Future, School of Biological Sciences, Monash University, Melbourne, Victoria, Australia, 8 Department of Zoology, University of Johannesburg, Johannesburg, South Africa, 9 Constantia Consulting, Christchurch, New Zealand, 10 Centre for Sustainable Ecosystem Solutions, School of Earth, Atmosphere and Life Sciences and Global Challenges Program, University of Wollongong, Wollongong, New South Wales, Australia, 11 Securing Antarctica's Environmental Future, University of Wollongong, Wollongong, New South Wales, Australia, 12 Centre d'Etudes Biologiques de Chizé, La Rochelle Université − CNRS, UMR 7372, Villiers en Bois, France, 13 Natural History Museum of Denmark, University of Copenhagen, Copenhagen, Denmark, 14 Antarctic and Southern Ocean Coalition, Washington DC, United States of America, 15 Centre for Microbial Ecology and Genomics, Department of Biochemistry, Genetics and Microbiology, University of Pretoria, Pretoria, South Africa, 16 University of Rennes 1, CNRS, EcoBio (Ecosystèmes, biodiversité, évolution)—UMR 6553, Rennes, France, 17 Biology Department, Woods Hole Oceanographic Institution, Woods Hole, Massachusetts, United States of America, 18 International Association of Antarctica Tour Operators (IAATO), South Kingstown, Rhode Island, United States of America, 19 Reel Time Gaming, Taringa, Queensland, Australia, 20 Department of Ecology and Evolution, Stony Brook University, Stony Brook, New York, United States of America, 21 Norwegian Polar Institute, Fram Centre, Tromsø, Norway, 22 Department of Biology, Universidad Autónoma de Madrid, Madrid, Spain, 23 Institute of Ecology and Evolution, University of Oregon, Eugene, Oregon, United States of America, 24 Viking Expeditions, Basel, Switzerland, 25 Faculty of Environment and Information Studies, Keio University, Fujisawa, Kanagawa Japan, 26 National Institute of Polar Research, Tachikawa, Tokyo, Japan, 27 Department of Biology and School of Global Environmental Sustainability, Colorado State University, Fort Collins, Colorado, United States of America, 28 InBios Research Unit, University of Liège, Liège, Belgium

* jasmine.lee1@uqconnect.edu.au



**Data Availability Statement:** Antarctic priority threat management database containing intactness values, benefits and uncertainties for each biodiversity taxon are available from the Australian Antarctic Data Centre (AADC; https://doi.org/10.

## Abstract

Antarctic terrestrial biodiversity faces multiple threats, from invasive species to climate change. Yet no large-scale assessments of threat management strategies exist. Applying a structured participatory approach, we demonstrate that existing conservation efforts are insufficient in a changing world, estimating that 65% (at best 37%, at worst 97%) of native terrestrial taxa and land-associated seabirds are likely to decline by 2100 under current trajectories. Emperor penguins are identified as the most vulnerable taxon, followed by other seabirds and dry soil nematodes. We find that implementing 10 key threat management strategies in parallel, at an estimated present-day equivalent annual cost of US$23 million,

26179/5da8f8e7a2256). Action, cost and feasibility information, as well as the numerical values underlying the Figures, are included within the paper and its Supporting Information files.

**Funding:** This project was supported by the Scientific Committee on Antarctic Research (SCAR), who provided support for the meeting, and by the Australian Antarctic Science Program (projects 4296, 4297 and Integrated Digital East Antarctica - IDEA). J.L. was supported by the Holsworth Wildlife Research Endowment – Equity Trustees Charitable Foundation, an Australian Government Research Training Program Scholarship, and a Research Fellowship from The Royal Commission for the Exhibition of 1851. P.C., K.H. and E.B. are supported by NERC core funding to the British Antarctic Survey 'Biodiversity, Evolution and Adaptation' Team (P.C., E.B.) and Environment Office (K.H.). E.B. was also supported by a NERC-CONICYT grant NE/P003079/1 and Carlsberg Foundation grant CF18-0267. A.W. is a Senior Research Associate of the FRS-FNRS and supported by the CDR J.0152.18 and BelSPO project BR/165/A1/MICROBIAN. S.J. was supported by NSF OPP 1840058 and 1744794. A. Q. was funded by Agencia Estatal de Investigación (Ministry of Science and Innovation) through the grant CTM2016-79741-R. The funders had no role in study design, data collection and analysis, decision to publish, or preparation of the manuscript.

**Competing interests:** The authors have declared that no competing interests exist.

**Abbreviations:** ATCM, Antarctic Treaty Consultative Meeting; ATCP, Antarctic Treaty Consultative Parties; CEP, Committee for Environmental Protection; COMNAP, Council of Managers of National Antarctic Programs; COVID-19, Coronavirus Disease 2019; FTE, full-time equivalent; IAATO, The International Association of Antarctica Tour Operators; ILP, integer linear program; IPCC, Intergovernmental Panel on Climate Change; NAP, National Antarctic Program; POP, persistent organic pollutant; PTM, priority threat management; PV, present value; RCP, Representative Concentration Pathway; SSP, Shared Socioeconomic Pathway; UNFCCC, United Nations Framework Convention on Climate Change.

could benefit up to 84% of Antarctic taxa. Climate change is identified as the most pervasive threat to Antarctic biodiversity and influencing global policy to effectively limit climate change is the most beneficial conservation strategy. However, minimising impacts of human activities and improved planning and management of new infrastructure projects are cost-effective and will help to minimise regional threats. Simultaneous global and regional efforts are critical to secure Antarctic biodiversity for future generations.

## Introduction

Conserving Antarctic species for future generations is in the interest of all humankind. Designated as a natural reserve, devoted to peace and science, Antarctica is home to numerous endemic species [1]. Charismatic emperor and Adélie penguins capture the world's imagination, while the nematode *Scottnema lindsayae* survives in saline soils that are inhospitable to other eukaryotic life forms [2]. These endemic species possess distinctive adaptations that allow them to survive Antarctica's extreme conditions [3,4]. Some of them may even be key for developing new technologies or medicines, including sustainable biomanufacturing processes, uses in the frozen food industry [5], and biodiesel production [6]. Furthermore, Antarctica and the Southern Ocean provide essential ecosystem services, in particular, regulating global climate via processes driving atmospheric circulation and ocean currents and the absorption of anthropogenic heat and $CO_2$ [7,8]. Finally, Antarctica is one of the few places on the planet that can still be considered largely unspoilt by the industrial development of humanity [9].

Although Antarctica is relatively free from many of the environmental threats that beset the rest of the world, e.g., deforestation to generate land for agriculture, threats to Antarctic biodiversity are intensifying at an unprecedented rate [10]. The Antarctic Peninsula was one of the most rapidly warming regions globally in the second half of the 20th century [7]. This trend has recently paused [11], but many locations have still experienced short-term extreme events, such as heat waves with record high air temperatures (18.3˚C; [12], or +40˚C above average; [13]), and recent studies report evidence of a strong warming trend re-establishing (e.g., on the South Orkney Islands; [14]). Scientific activities and associated infrastructure are expanding [15] and annual tourist numbers have increased more than 8-fold since the 1990s, to nearly 75,000 in 2019/2020 [16,17], although then experiencing a temporary hiatus as a result of the global Coronavirus Disease 2019 (COVID-19) pandemic [18]. The current suite of Antarctic protected areas does not represent the continent's full range of biodiversity and some are experiencing anthropogenic pressures [19–22]. Although Antarctica's geographic isolation and extreme climate have historically afforded some protection to the continent, the combination of increasing human activity and warming is also lowering the barriers to the arrival and establishment of non-native species [23–25].

The threats to Antarctic species and ecosystems are increasingly well documented; however, species vulnerability to threats is poorly understood, and decision-makers also lack the information required to prioritise, develop, and implement threat management responses [10]. Understanding how vulnerable species are to threats, and how varying conservation actions would benefit those species, is important for tailoring appropriate conservation responses across time and space [26,27]. A number of studies examine vulnerability of a particular species or taxa to a specific threat (e.g., open top chamber experiments indicate the springtail *Cryptopygus antarcticus* is vulnerable to warming; [28]), though there have been limited efforts

to undertake broad scale quantifications of vulnerability across terrestrial groups. Species that are not vulnerable are less likely to require investment of conservation resources.

Resources for conservation are often finite and it is important to understand how and where time and effort should be invested to achieve the best outcomes [29,30]. Cost-effectiveness approaches have been demonstrated to provide conservation solutions that create greater benefits for biodiversity with limited resources, compared to when cost-effectiveness is not considered [31–33]. Identifying the conservation strategies that provide the highest expected benefit can also be useful when cost estimates are uncertain or when cost is not a barrier [32]. Despite the interest in conserving Antarctic biodiversity, there is no comprehensive assessment of the costs of effective conservation in Antarctica, or the expected benefits and cost-effectiveness of applying various conservation strategies. Some specific examples exist, e.g., Raymond and Snape [34], who use triage to identify potential candidate sites for undertaking remediation.

Antarctic Treaty Parties recognise the importance of drawing on the best available science to inform decisions, yet key questions regarding where and how conservation resources should be invested remain unanswered. Here, we seek to answer 3 of the most pressing questions: (1) Which terrestrial Antarctic taxa are most vulnerable to threats? (2) What management actions and resource investment will provide the greatest benefit to biodiversity and ensure persistence across all terrestrial biodiversity groups? (3) Which management strategies should we focus on to provide the highest conservation return on investment? We answer these questions by undertaking a comprehensive quantification of the relative importance and return on investment of existing and potential management actions and strategies for reducing threats and securing Antarctica's biodiversity. We also provide crucial first estimates of the costs of these strategies.

## Our approach

Antarctica's size, isolation, and extreme environmental conditions make research challenging. Our understanding of Antarctica's biodiversity is hampered by a lack of quantitative data, especially on species interactions, taxonomy, and baseline data on abundance and distributions. In this context, expert knowledge, together with available empirical data, can significantly improve the foundation on which conservation plans and management actions are based [35]. Antarctic experts have accumulated decades of experience in the region and hold unique knowledge encompassing most terrestrial biodiversity groups. Using this knowledge, we apply a structured, participatory decision-science approach [36,37] to quantify and prioritise cost-effective and complementary strategies for achieving biodiversity conservation in terrestrial Antarctica (see Materials and methods).

First, experts collectively defined a set of management strategies with the aim of safeguarding Antarctic biodiversity until 2100 (Table 1), for both the Antarctic Peninsula and continental Antarctic regions (see Fig 1B). It was important to consider these regions separately because they are climatically and biologically distinct [1], and the threats and the species responses to threats are likely to be different between the 2 regions. Each management strategy was targeted towards abating local threats (e.g., remediating contaminated sites), except for the "Influence external policy" strategy that was unique in being targeted towards reducing global threats (namely climate change). Each management strategy is made up of multiple conservation actions which, when implemented together, will achieve the strategy's objectives (Table 1). Experts estimated the cost and feasibility for undertaking each action (see Materials and methods). For example, developing a best practice manual for remediation of contaminated sites was estimated to require 1 full-time equivalent (FTE) employee for the first year

**Table 1. Overview of proposed management strategies for conserving Antarctic biodiversity.**

| Strategy name | Objectives and details |
|---|---|
| Business as usual (Baseline) | Continue with actions and strategies currently in use, but neither expand on these strategies nor employ new strategies. Baseline against which to measure other strategies. |
| Remediation (Remediate) | Increase amount of, or improve, quality of habitat available to biodiversity in comparison to habitat currently available. By remediating 20 environmentally damaged (physically, chemically, biologically) sites (including freshwater) that will provide the greatest benefit to biodiversity, including remediation of legacy waste sites if necessary. |
| Manage existing infrastructure (Exist infra.) | Reduce and minimise impacts of existing infrastructure compared to current levels. |
| Manage new infrastructure (New infra.) | Prevent, reduce, and minimise impacts of new infrastructure. |
| Transport management (Transport) | Reduce and minimise impacts of transport compared to current levels. |
| Manage non-native species and disease (Non-native) | Reduce impacts of non-native species and disease on native biodiversity. Where possible prevent establishment of new populations of non-native species. Eradicate or, if not possible, minimise impacts of established non-native species. |
| Protect vegetation from physical impacts (Protect vegetation) | Reduce physical impacts of human activities and native vertebrate activities on vegetation. Halt the decline (or loss) of vegetation and associated taxa due to direct physical damage/impact at key sites in the Antarctic Peninsula (e.g., animal damage). |
| Protecting areas (Protecting areas) | Reduce impacts of human activities on biodiversity by increasing the amount and representation of habitat in protected areas. Develop the ASPA system to improve representation of the values specified in the Environmental Protocol and ensure the network incorporates contemporary systematic conservation planning pillars. |
| Managing and protecting species (Protecting species) | Reduce threatening impacts on taxa identified as threatened by 2100. 1. Identify and protect threatened species (assume 10–15 species to be identified and listed under this strategy). 2. Prevent extinction of native species in situ. |
| Minimise impacts of human activity (Human activities) | Prevent and minimise physical impacts on biodiversity and habitats compared to current levels that stem from human activities in Antarctica (e.g., fieldwork, tourism, station activities). Relevant where improving education and training, and implementing standard practices, on-ground operating procedures and compliance is likely to reduce impacts through changes in human behaviour. |
| Influence external policy (Influence ext. policy) | Minimise or reduce impacts of threats (primarily climate change) on Antarctic biodiversity that originate externally via engagement with appropriate policy bodies and raising public awareness. **Note:** Outcome of strategy is an assumption that the Paris Climate Agreement of <2°C warming is achieved. Use RCP2.6 instead of 4.5/8.5. |
| All strategies excluding "influence external policy" (All strats. excl IEP) | All management strategies combined except "Influence external policy." |
| All strategies combined (All strategies) | All management strategies combined. |

Shorthand strategy names are given in brackets. "Current" refers to the state of the Antarctic environment and associated management actions in 2017. Grey shading identifies global strategies or a combination of regional and global strategies. More details on specific actions, costs, and feasibility are provided in S1 Data.

ASPA, Antarctic Specially Protected Area.

and an ongoing 0.25 FTE each year to update and maintain the manual, with a likelihood of uptake of 100%. This is the equivalent of 21.75 FTE and would cost $1.7 M over the 83-year timeframe. The conservation actions, and cost and feasibility of each action, are detailed in S1 Data. Because the "Influence external policy" strategy is targeted toward reducing global

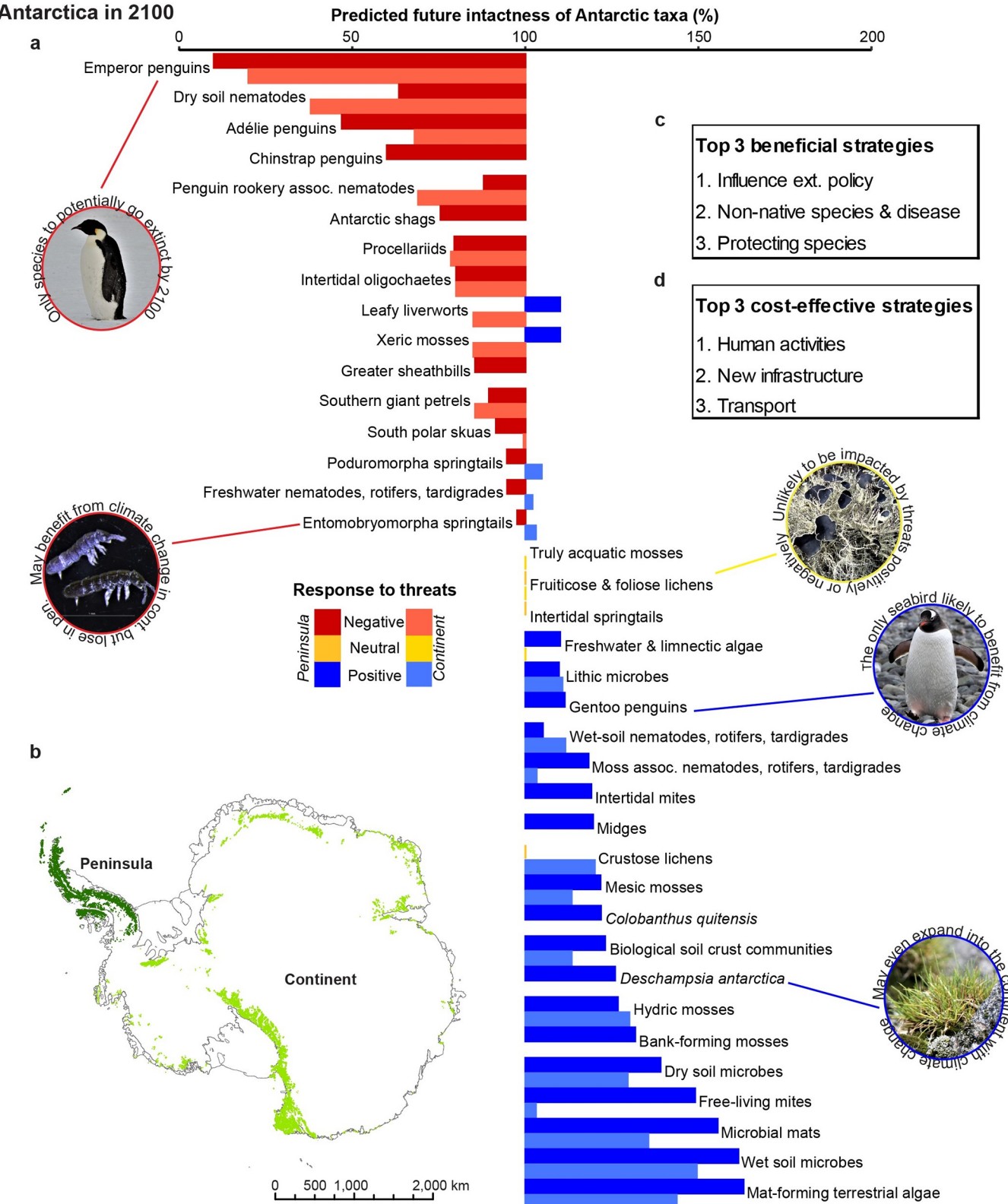

**Fig 1. Vulnerability of terrestrial Antarctic biodiversity to all threats under climate forcing scenario RCP8.5 and the most beneficial and cost-effective conservation management strategies. (a)** Regional vulnerability of biodiversity groups to all potential threats, where colours represent each taxon's expected response to threats, with darker/lighter shadings denoting the regional delineation as peninsula or continent, respectively. Bars represent experts' best estimate

of the future intactness of each taxon relative to current (100%) intactness if no additional conservation strategies are implemented. Taxa with values below 100% are predicted to be vulnerable, while taxa with values beyond 100% are predicted to benefit. (**b**) The 2 main Antarctic regions considered in this study. (**c**) The top 3 individual management strategies that would provide the highest total benefit to biodiversity. (**d**) The top 3 most cost-effective strategies for conserving biodiversity. The data underlying this figure can be found in S2 Data. The Antarctic coastline file for the map has been downloaded from the Antarctic Digital Database (ADD Version 7; http://www.add.scar.org).

threats, while we identified costs and actions, they are focused only on the tasks that can be undertaken directly by the Antarctic community (i.e., public and policy engagement; see S1 Data). In consequence, the estimated costing does not represent the full global cost (nor the full global benefits) of reducing emissions to limit warming to 2°C [38] and we therefore did not include "Influence external policy" in the cost-effectiveness or complementarity analyses.

Conservation benefits were quantified at the strategy level, where the benefit was estimated using the difference between current and predicted future levels of intactness of biota. We defined intactness as a measure of how "intact" a taxon will be compared to 2017 levels (see Materials and methods). A score of 100 indicates—"intactness same as today," 0 - "taxon is completely degraded," and 200 - "taxon is doing twice as well." A metric based on extinction risk or persistence probability was considered inappropriate for Antarctic biodiversity given that very few species are predicted to be at risk of extinction by the end of the century. As per expert elicitation protocols [39], biodiversity experts provided a best estimate, and lower (worst-case scenario) and upper (best-case scenario) bounds of predicted future intactness of each taxon in response to each management strategy and for a "business as usual" baseline (Table 1). They also provided a confidence estimate to capture uncertainty, which together with the bounds were used in a sensitivity analysis [37,40]. They also identified knowledge shortfalls for each taxon (Tables A and B in S1 Text; [41]).

Using the cost, feasibility, and benefit estimates we assessed the expected benefit, cost-effectiveness, and complementarity of strategies for conserving Antarctic biodiversity to the end of the 21st century under Intergovernmental Panel on Climate Change (IPCC) climate change scenarios [42]. For the "Influence external policy" strategy, we assessed the option of influencing global climate policy to abate the threat of climate change, where an outcome of the successful implementation of this strategy is that greenhouse gas emissions are reduced sufficiently to meet the 2°C Paris Climate Agreement, limiting warming to <2°C above pre-industrial levels and ideally to 1.5°C [38]. Thus, "Influence external policy" was assessed under the Representative Concentration Pathway (RCP) 2.6 scenario adopted by the IPCC, the only RCP scenario that keeps global warming to <2°C [42,43]. All other strategies were assessed under RCP4.5 and RCP8.5, where RCP4.5 represents moderate carbon emissions and RCP8.5 a more severe scenario [42].

## Results and discussion

### Threat vulnerability

Taxa varied in their expected vulnerability to threats by 2100 (Fig 1A, see raw values on the Australian Antarctic Data Centre; https://doi.org/10.26179/5da8f8e7a2256). Some species are predicted to decline under future conditions, while others are expected to benefit, with climate change likely to be the primary factor driving these responses. Emperor penguins (*Aptenodytes forsteri*) were identified as the most vulnerable taxon, and the only one at risk of possible extinction (see worst-case scenario/lower bound: https://doi.org/10.26179/5da8f8e7a2256), followed by dry soil nematodes and Adélie (*Pygoscelis adeliae*) and chinstrap (*P. antarcticus*) penguins. Recent declines in dry soil nematodes in the McMurdo Dry Valleys have been linked to a changing climate [44], as have declines among Adélie penguins in the Atlantic sector [45].

Up to 80% of emperor penguin colonies are projected to be quasi-extinct by 2100 (decline by >90%) with business-as-usual increases in greenhouse gas emissions [46]. However, if Paris Climate Agreement measures to limit warming to <2˚C are met, this estimate reduces to 31% [46]. Several taxa are predicted to be vulnerable in only 1 region while increasing intactness in the other (e.g., xeric mosses), though the taxa may be classed as vulnerable overall when averaged. Research targeted to better understand why some taxa are more vulnerable than others (i.e., sensitivity and exposure; [47]) and the spatial patterns of these drivers across taxa, can inform conservation actions for particular taxa or regions [26]. Some of the most vulnerable taxa may also warrant special protection through designation under Annex II of the Environmental Protocol as Specially Protected Species, as has already been suggested for emperor penguins [48].

Some taxa were not predicted to be sensitive to climate change or other threats and their intactness is expected to remain similar to current levels. Others may expand their distribution and/or abundance due to future changes in the region, including those that are primarily constrained by climate-related abiotic factors such as temperature and limited water availability [4]. Indeed, evidence suggests this is already occurring—the increase in growth rate of bank-forming moss on the Antarctic Peninsula [49], rapid expansion in some populations of the 2 native flowering plant species *Colobanthus quitensis* and *Deschampsia antarctica* [14,50], and southward range extensions of gentoo penguins (*P. papua*; [45]). However, these species will not continue expanding indefinitely and the longer-term impacts of environmental change for all species remain uncertain and complex. The expansion of some taxa may also negatively impact other native species at a competitive disadvantage [51]. Some experimental evidence suggests that the invasive grass *Poa annua* is likely to compete with, and possibly outcompete, the 2 native flowering plants [52,53].

## Benefits of management strategies

Under the current management scenario and if global greenhouse gas emissions continue to track above 2˚C warming [54], approximately 65% (at best 37% and at worst 97%) of taxa will decline by 2100 (Fig 2A—"Baseline;" Fig A in S1 Text). These results demonstrate that current conservation actions under the provisions of the Protocol on Environmental Protection to the Antarctic Treaty are insufficient to conserve Antarctic biodiversity in a warmer future. If comprehensive management is implemented ("All strategies combined" approach), approximately 63% (minimum 37%, maximum 84%) of taxa will benefit and the number of taxa declining compared to their current intactness could be reduced to 42% (Fig 2A). However, 53% to 61%, and up to 68%, of taxa would benefit if the single "Influence external policy" strategy aimed at reducing global climate change was successful. If climate change cannot be mitigated, all regional strategies combined ("All strategies combined excluding policy influence") will still benefit 53% to 55%, and up to 74% of taxa (Fig 2A), though the total amount of benefit achieved is reduced (Fig 2B). The individual strategy benefitting the next most taxa is "Manage non-native species and disease" (47% for both RCP4.5 and RCP8.5), and then both "Managing and protecting species" and "Protecting areas" for RCP4.5 (45%) and "Protecting areas" for RCP8.5 (40%).

The strategies that provide the highest total expected benefits, when combined for all taxa and both regions, are almost the same regardless of climate forcing scenario (Fig 2B and Table 2). "All strategies combined," unsurprisingly, provides the highest total benefit. The second highest total benefit is "Influence external policy" followed by "All strategies combined excluding policy influence." For RCP4.5, the next best strategy is "Managing and protecting species," while for RCP8.5, it is "Manage non-native species and disease" (Fig 2B and Table 2).

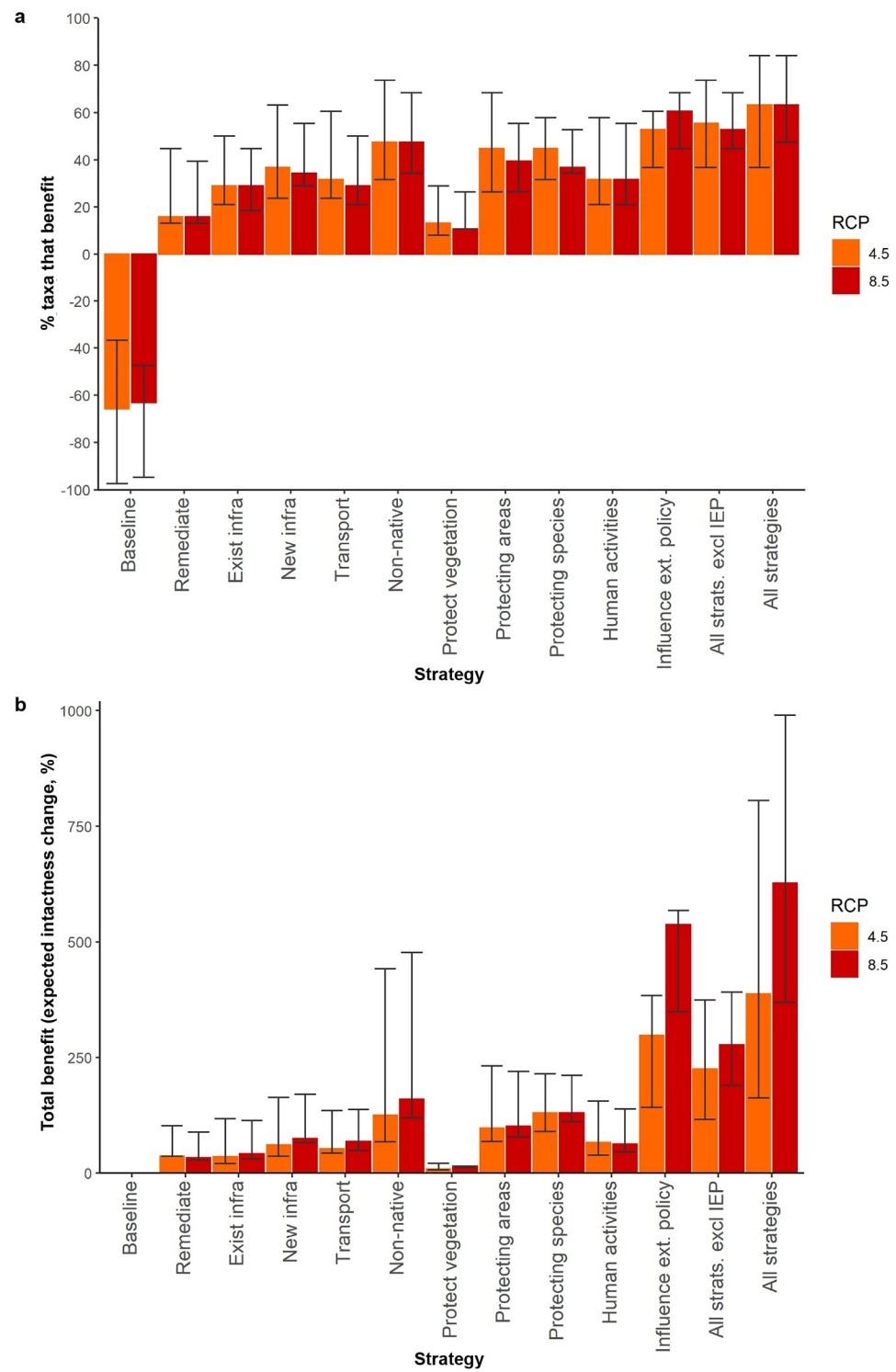

**Fig 2. Response of Antarctic terrestrial biodiversity to various conservation management strategies by the end of 2100 under 2 climate forcing scenarios (RCP4.5, RCP8.5).** (**a**) Percentage of taxonomic groups likely to benefit. (**b**) Total expected benefit of strategies summed for all taxa and both regions combined. Bars represent the experts' best estimates when assessing benefit, while error bars represent upper (best-case scenario) and lower (worst-case scenario) bounds. An outcome of the "Influence external policy" (IEP) and "All strategies combined" strategies is that carbon emissions are reduced globally (in line with the milder RCP2.6); however, benefits are still calculated relative to the baselines of RCP4.5 and RCP8.5. Values used to calculate benefit were capped at current (100%) intactness (see Fig A

in S1 Text for inclusion of benefits beyond current intactness). The data underlying this figure can be found in S2 Data. RCP, Representative Concentration Pathway.

Regionally, the same 3 strategies deliver the greatest benefit, though higher benefits are generally predicted for the Antarctic Peninsula region than the continent (Fig B in S1 Text), possibly due to the greater projected changes there.

While the above summarises the number of taxa that would benefit and the total amount of benefit predicted under each management strategy for all taxa, it does not provide information on the amount of benefit to an individual taxon (where some taxa may benefit far more from a management strategy than others). Responses to management strategies vary among taxa in both scale and direction (see benefits for individual taxa shown on the Australian Antarctic Data Centre; https://doi.org/10.26179/5da8f8e7a2256), for example, Adélie penguins are predicted to increase intactness by approximately 45% with the "Influencing external policy" strategy (RCP8.5), while Antarctic shags are predicted to benefit by approximately 10%. Most strategies reduce predicted declines. However, for those species that are predicted to benefit with climate change, at least initially (e.g., gentoo penguins), they benefit less with the milder RCP2.6 scenario in contrast to RCP4.5 or RCP8.5. Thus, it appears that they have reduced intactness with the RCP2.6 "Influence external policy" strategy compared to the baseline RCP4.5 or RCP8.5 values.

### Feasibility and costs

The relationships between expected benefit, estimated cost, and feasibility are illustrated in Fig C in S1 Text. "All strategies combined" and "All strategies combined excluding policy influence" are both high cost, high benefit strategies (Fig Ca in S1 Text), whereas "Influence external policy" is low cost (in terms of direct cost to Antarctic decision-makers and managers), high benefit (Fig Ca in S1 Text). However, "Influence external policy" has the lowest feasibility of all strategies, where the experts estimated it to have a "likelihood of success" of 5% in the short term and 45% from year 30 onwards (averaged at 12% over the 83 years when

**Table 2. Evaluation of key management strategies for Antarctic biodiversity until 2100 under 2 climate forcing scenarios (RCP4.5, RCP8.5).**

| Strategy | Cost (US$ M) | Feasibility | RCP4.5 | | RCP8.5 | |
|---|---|---|---|---|---|---|
| | | | Benefit | CE rank | Benefit | CE rank |
| Minimise impacts of human activity | 11.48 | 0.31 | 66.0 | 1 | 62.1 | 1 |
| Manage new infrastructure | 31.81 | 0.60 | 60.5 | 2 | 74.3 | 2 |
| Transport management | 39.08 | 0.72 | 52.3 | 3 | 68.3 | 3 |
| Protecting areas | 99.82 | 0.70 | 96.7 | 4 | 101.1 | 4 |
| Managing and protecting species | 215.94 | 0.53 | 129.6 | 5 | 130.2 | 5 |
| Remediation | 109.85 | 0.54 | 35.8 | 6 | 32.9 | 7 |
| Protect vegetation from physical impacts | 31.72 | 0.37 | 8.3 | 7 | 13.8 | 6 |
| Manage non-native species and disease | 762.62 | 0.58 | 124.9 | 8 | 159.3 | 8 |
| All strategies excl influence ext. policy | 1923.84 | 0.56 | 224.7 | 9 | 277.3 | 9 |
| Manage existing infrastructure | 811.43 | 0.60 | 34.8 | 10 | 40.8 | 10 |
| Baseline | 0 | 1.00 | 0 | 11 | 0 | 11 |

Including estimated total cost over the next 83 years (PV; using social discount rate of 2%), estimated feasibility, total expected benefit (% change combined for all taxa), and cost-effectiveness rank. Ranked in order of CE for RCP4.5. Benefits of strategies are calculated using best estimates of improved intactness provided by experts (see Table E in S1 Text for calculations using upper and lower bounds) and were capped at current (100%) intactness.

CE, cost-effectiveness; PV, present value; RCP, Representative Concentration Pathway.

incorporating the "likelihood of uptake" estimates), reflecting the uncertainty around the sociopolitical dynamics of climate action and also noting global governments' so far unsuccessful efforts to drastically reduce carbon emissions [55]. For the other strategies, "Minimise impacts of human activities" has the second lowest feasibility. "Transport management" and "Protecting areas" have the highest feasibility after the baseline (Table 2 and Fig Cb in S1 Text).

Combining all strategies was consistently estimated to be the most expensive strategy, closely followed by "All strategies combined excluding policy influence" (Fig Da in S1 Text). Implementing all conservation strategies combined excluding "Influence external policy" is estimated, using a 2% discount rate (where future costs are discounted to present day values; PVs), to cost an average of US$23 million annually until 2100 if costs were borne evenly across years (in practice, a higher investment would be needed in earlier years and less in later years). In total, this equates to approximately US$1.92 billion (US$3.69 billion with 0% discount; Fig Da in S1 Text). This equates to 0.004% of the global GDP in 2019 (US$87.75 trillion; [56]), which is a comparably small investment in the context of (1) the timeframe and large number of nations committed to protecting the Antarctic environment; and (2) the future benefits to Antarctica's terrestrial and seabird biodiversity. However, determining how strategies will be funded is not straightforward due to Antarctica's governance arrangements, and costs would need to be shared across Treaty Parties and National Antarctic Programmes.

The third and fourth most expensive strategies were "Managing existing infrastructure" and "Managing non-native species and disease," respectively. Both managing non-native species and managing and protecting individual species were estimated to be relatively expensive (approximately US$763 and $216 million, respectively) as they include baseline biodiversity surveys and could require intensive on-ground action, such as translocating individuals (e.g., as has been done for moss beds on King George Island; [57]) or eradicating non-native species (S1 Data). Yet if climate change was reduced or largely prevented, the risk of non-native species invasion and necessary management of individual species is likely to be reduced in comparison to what would otherwise be necessary due to the synergistic effect of climate change on non-native species survival and expansion [25,58,59]. Some established non-native species are, however, already expanding their ranges [60,61]. Consequences of milder climates, such as reduced sea ice extent, may also increase accessibility for science and tourism, thereby increasing the potential inadvertent transfer of non-native species. The cheapest strategy was "Minimise impacts of human activity" (approximately US$11.5 million), followed by "Influence external policy" (approximately US$14 million), and "Protect vegetation from physical impacts" (approximately US$32 million). However, as noted above, the "Influence external policy" strategy does not include the global costs of reducing emissions, and thus, we have excluded it from the cost-effectiveness and complementarity analyses below.

Regionally, it would be more expensive to implement management strategies in the Antarctic Peninsula (Db in S1 Text). This is due to rapid environmental change and the concentration of human activity in the region resulting in a larger effort needed for strategies to be successful (Table C). The largest cost components varied depending on the strategy (see S1 Data), but overall, the major elements were funding long summer Antarctic berths (living costs and transport), followed by the costs of Antarctic FTE salaries (Fig Dc in S1 Text).

## Cost-effective strategies

Strategies varied in their cost-effectiveness, though the highest ranked strategies were consistent across regions and RCP's (Tables 2 and Da in S1 Text), demonstrating our analyses are robust to alternative future climate warming scenarios. Several strategies were identified as

providing the highest conservation return on investment, specifically: minimising impacts of human activities, improved planning and management of new infrastructure projects, and improving transport management (e.g., reducing pollution and disturbance caused by vessels and aircraft; Table 2). Regionally, and with analyses using the upper and lower bound estimates, the same 3 strategies were consistently ranked as the most cost-effective, though their order varied (Tables D and E in S1 Text). Two of these 3 strategies ("Minimise impacts of human activity" and "Manage new infrastructure") were also identified as complementary (Fig 3). These strategies offer opportunities for rapid and cost-effective improvements that can support the more expensive, and already prioritised, management of non-native species [10]. For individual taxa, cost-effectiveness ranking of strategies varied (Tables F–I in S1 Text), indicating that several strategies that rank lower for biodiversity overall may be key for maintaining intactness in some taxa or locations, such as "Protect vegetation from physical impacts" for bank-forming mosses in the Antarctic Peninsula (Tables H and I in S1 Text).

## Complementary strategies

Sets of complementary strategies, which benefit as many taxa as possible under a given budget, varied depending on the intactness threshold and climate forcing scenario (Fig 3). However, generally the most cost-effective strategies were also identified as complementary. Combinations of "Minimise impacts of human activity," and "Manage new infrastructure" were often identified for smaller budgets of less than US$35 million, while "Protecting areas" was incorporated with an increased budget of around $100 million and "Managing and protecting species" with a budget of $216 million. If funding is not a barrier, then the $1.92 billion "All strategies combined excluding external policy" will maximise the number of intact taxa in some scenarios (Fig 3). Our analysis also identified 8 taxa that, in some scenarios (e.g., a 90% threshold under RCP8.5), are unable to reach the target intactness thresholds and may require increased attention from policymakers (Fig 3). These taxa include Adélie penguins, chinstrap penguins, Antarctic shags (*Leucocarbo bransfieldensis*), dry soil nematodes, emperor penguins, leafy liverworts, penguin-rookery associated nematodes, and xeric mosses. However, most taxa will retain a 70% intactness threshold even under the baseline, and all taxa can reach a 70% threshold under RCP4.5 with the implementation of several management strategies ("Minimise impacts of human activities," "Manage new infrastructure," and "Managing and protecting species") highlighting the lower predicted threats to Antarctic biodiversity in comparison to many regions globally [62], at least in this century. Given the generally lower abundance and the smaller growth windows for many Antarctic species [3], any reduction in intactness is considered substantial though.

## Implementation of threat management strategies

Climate change is the key threat to Antarctic life and it is clear that substantial conservation benefits could be achieved if climate change were reduced to less than 2°C warming, in line with the Paris Climate Agreement. Further limiting the warming trajectory to 1.5°C will likely provide even greater benefits, as predicted for emperor penguins [46,63] as well as biodiversity globally [64]. This highlights an urgent need to address the significant threats to Antarctic biodiversity arising from activities outside the Antarctic region, in particular climate change. Antarctic Treaty Consultative Parties (ATCPs) could proactively address this through enhanced engagement with relevant government bodies and intergovernmental processes (Antarctic Treaty, Article III.2, 1959), such as the IPCC and the United Nations Framework Convention on Climate Change (UNFCCC), to highlight the Antarctic implications of climate change and inform and support global climate action [65,66]. Climate change and associated

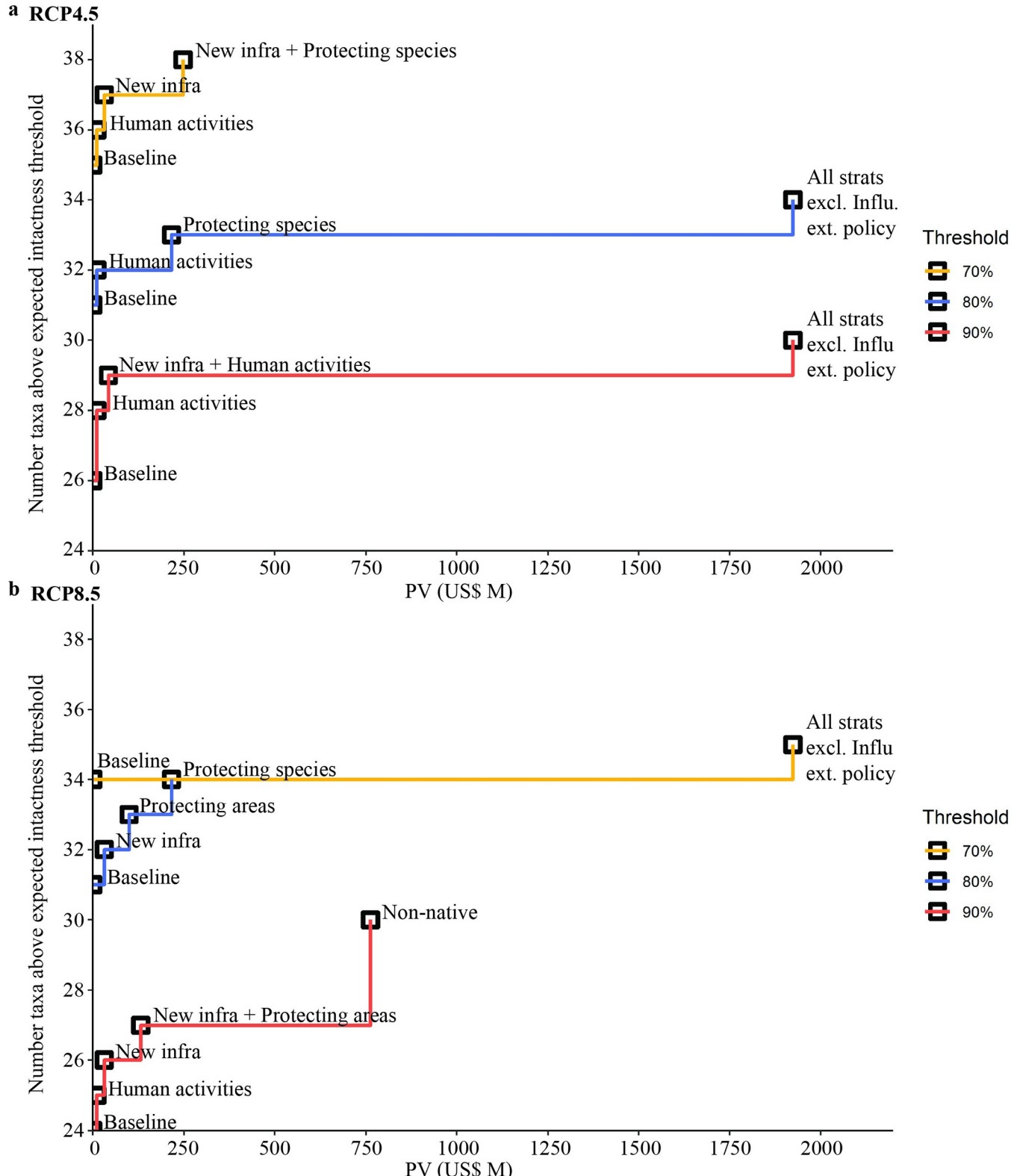

**Fig 3. Complementary solutions for conserving Antarctic terrestrial biodiversity for any given budget under 3 intactness thresholds and where there is no possibility of reducing climate scenario to the milder RCP2.6 through implementation of the "Influence external policy" strategy.** (a) RCP4.5 climate forcing scenario. (**b**) RCP8.5 climate forcing scenario. The steps represent the optimal strategies to invest in to ensure the maximum number of taxa possible reach an intactness threshold under any given budget. For example, if a budget of $250 M were available under RCP8.5, then the optimal strategy to invest in

for an 80% threshold is "Managing and protecting species," while for a 90% threshold it is "Manage new infrastructure" and "Protecting areas." Strategy names used here are abbreviated, and abbreviations are given in Table 1. Budget (over 83 years) is given as PV, where costs are discounted to equivalent present-day 2017 values using a 2% discount rate. Values used to calculate benefit, used in complementarity analysis, were capped at current (100%) intactness (An1). The data underlying this figure can be found in S2 Data. PV, present value; RCP, Representative Concentration Pathway.

environmental implications continue to be a major point of discussion at recent Antarctic meetings (e.g., [67] at ATCM XLIV—CEP XXIV), and ACTPs are well placed to represent these implications more broadly across other fora. Parties to the UNFCCC need to strengthen efforts to respond to the threat of climate change by meeting their obligations under the Paris Climate Agreement. A global threat cannot be abated by 1 region alone, and all regions and sectors have a role to play in addressing climate change. Antarctica provides essential ecosystem services for the entire planet and humanity cannot afford to ignore potential impacts to these systems [8]. Mitigating climate change will help to stabilise these Antarctic processes, which drive global ocean and atmospheric circulation, and which will play a critical role in future global sea-level rise [8,68]. Anthropogenic impacts on any of these processes could fundamentally change life on Earth [8].

Although we found that working towards global climate action will provide the largest conservation benefit and will help to ensure persistence of Antarctic terrestrial biodiversity, it is also essential to continue advancing Antarctic-centric conservation measures [10,69]. There is now an opportunity for ATCPs, and other stakeholders, to rapidly implement the most cost-effective and complementary regional strategies. In comparison to other regions globally (e.g., meeting restoration targets in the Brazilian Atlantic Forest hotspot is estimated to cost upwards of US$20 billion; [70], or recovering Australia's threatened species is likely to cost more than US$1.2 billion/year; [71]), the costs of implementing regional conservation actions in the Antarctic are relatively modest and should receive increased international attention. Minimising the impacts of human activity through influencing behaviour change and improving technology could provide substantial benefit to native species at an estimated cost of only US$11.5 million. Actions such as improving education for all visitors, optimising fieldwork to minimise research footprint, and developing and utilising remote sensing technology where possible, to reduce the number of field parties, will help to reduce impacts on biodiversity and habitats. Improved planning for and management of new infrastructure (approximately US$32 million) can help to prevent and reduce unnecessary impacts of National Antarctic Programs (NAPs). Carefully considering the locations and the need to establish such infrastructure is essential [10,15]. National operators could also commit to further sharing infrastructure and logistics where possible [15] to reduce both carbon and spatial footprint, as well as costs [72]. The initial success of a logistic-sharing trial in the Antarctic Peninsula, organised by the Council of Managers of National Antarctic Programs (COMNAP), demonstrates the potential of such approaches [73], though noting that the use of some infrastructure is already oversubscribed. Improving transport management by better utilising transport infrastructure and technology to reduce transport or black carbon pollution [74,75], and by optimising routes and timing to minimise wildlife disturbance [76] is another feasible and low-cost strategy. The International Association of Antarctica Tour Operators (IAATO) has commenced taking steps in this direction by establishing 2 "go-slow" whale zones in the Antarctic Peninsula region [77].

Simultaneously, ATCPs could prioritise further efforts toward enacting some of the more ambitious conservation strategies, with a focus on providing the highest benefits to biodiversity. Managing and protecting threatened species, while costly, is expected to deliver substantial benefits to these species through designation as Specially Protected Species [78] and by

developing and implementing recovery plans and on-ground management (including ex situ conservation, if necessary). Improving management of non-native species and disease by preventing new establishment events and eradicating or, if that is not possible, managing established populations is increasingly important to reduce a threat that is likely to act synergistically with climate change [4,24,25,69]. Currently, national programs differ substantially in their implementation of environmental management strategies (such as the degree of biosecurity for vessels and staff; [24,60]), and in some cases, there may be large benefits in promoting standard (or at least compatible) levels of conservation management across programs. COMNAP has an opportunity to make further substantial contributions toward resolving this issue.

## Uncertainty

Expert-based assessments are valuable for informed and precautionary decision-making in Antarctica, where there is often a lack of comprehensive quantitative data. This is highlighted by the >30 taxa for which the experts identified Eltonian (lack of knowledge on ecological interactions) and/or Prestonian (lack of understanding of species abundance and population dynamics) knowledge shortfalls ([41]; Fig 4; shortfall definitions in Table A in S1 Text; shortfalls for individual taxa in Table B in S1 Text). Utilising the collective knowledge of experts to help bridge these gaps, as we did here, is a useful and appropriate alternative, though uncertainty also arises from expert assessments [39,40,79,80]. In some studies, experts have also been found to be overconfident and to underestimate the uncertainty around their estimates [80,81], which should be considered in interpretation of results.

We captured some of the uncertainties of the expert's assessments by eliciting a range of lower to upper bounds for expected benefit (e.g., Fig 2) and in the confidence estimates. Utilising these estimates in sensitivity analyses helped us to determine whether results are robust in light of expert's uncertainties [37,40,82]. A sensitivity analysis of the cost, feasibility, and benefit values provided by the experts indicated that the strategies cost-effectiveness rankings were reasonably robust to all sources of uncertainty considered here (see Figs E and F in S1 Text). While there was some overlap of the most pessimistic bounds of the optimal strategies with the most optimistic bounds of the mid-level strategies (e.g., if the most optimistic bound of "Protecting areas" and the most pessimistic bound of "Managing transport" represented true values, then "Protecting areas" would be the more cost-effective of the 2), the same strategies were still consistently highlighted as the most cost-effective (Figs E and F in S1 Text). Thus, despite the uncertainties in expert's estimates, our results suggest the top 3 most beneficial ("Influence external policy," "Manage non-native species and disease," "Managing and protecting species"), or most cost-effective strategies ("Minimise impacts of human activity," "Manage new infrastructure," "Transport management") are the same regardless of RCP, region, or whether upper or lower bound is used (see Table 2 and Figs E and F and Tables D and E in S1 Text). Future research targeting the identified knowledge shortfalls (Table B in S1 Text), as well as increased accessibility of cost and feasibility information for implemented Antarctic management options, could help to reduce data gaps and uncertainty in future analyses [83].

A lack of empirical data need not hinder conservation decisions and the implementation of management strategies, especially in an era of rapid global change [35,84–86]. The cost-effectiveness approach employed here provides robust decision support based on expert judgement. Acknowledging uncertainties is important but deferring decisions while filling quantitative data gaps can result in worse conservation outcomes [84,87]. Making no decision is still a decision (as represented here by the baseline "Business as usual" strategy), and our results

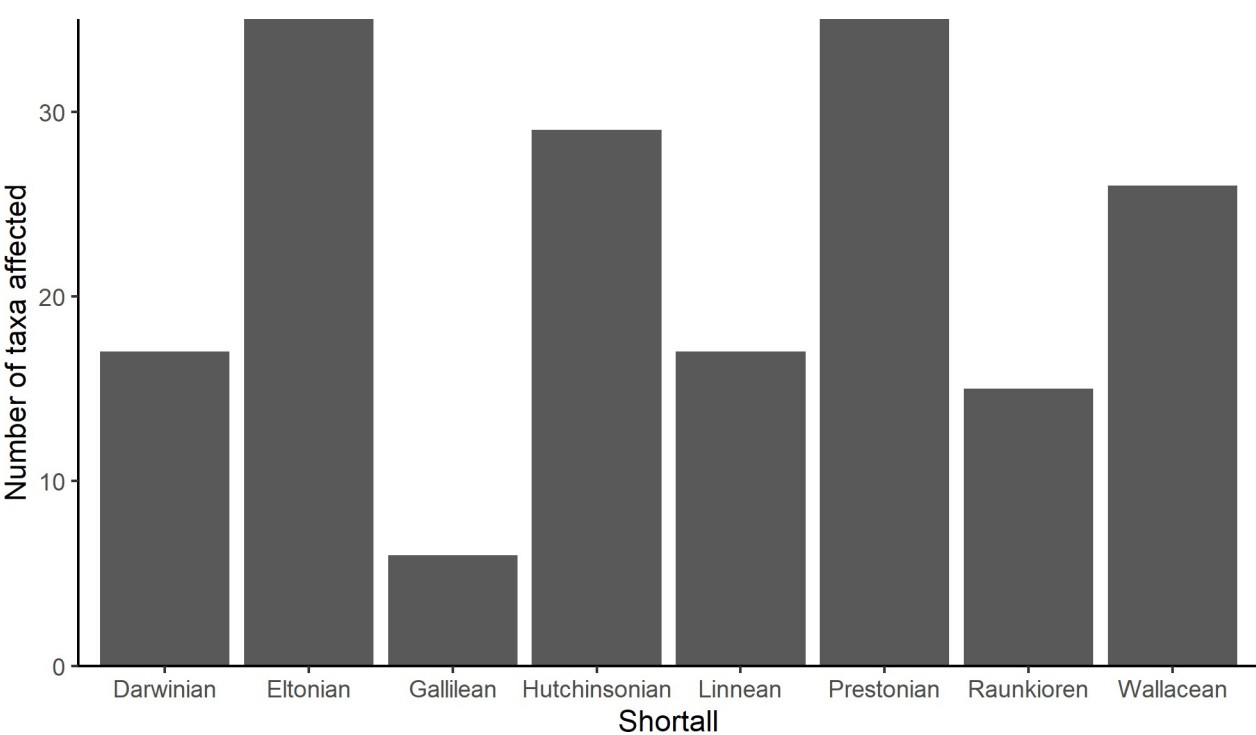

**Fig 4. Number of taxa for which each of 8 knowledge shortfalls were identified that limit understanding and assessment of Antarctic terrestrial biodiversity.** See Table A in S1 Text for a definition of each of the 8 shortfalls and Table B in S1 Text that lists the shortfalls identified for each biodiversity taxon individually. The data underlying this figure can be found in S2 Data.

unequivocally suggest that continuing with business-as-usual will lead to declines for some Antarctic taxa. Antarctic Treaty Parties must continue to advance Antarctic conservation using the best science available today.

## Conclusions

Conserving Antarctic biodiversity for future generations requires political commitment and practical action both within the Antarctic region and globally. Our analysis identifies and prioritises the management strategies and resources required to promote the continued persistence of Antarctic terrestrial biodiversity and represents a significant advance in the "best available science" to inform decision-making and management. The ATCPs reaffirmed their commitment to comprehensive protection of the Antarctic environment and dependent and associated ecosystems through the Paris Declaration of 2021 (Environmental Protocol, Article II, 1991; [88,89]). Parties reaffirmed their commitment "to take account of best available scientific and technical advice in the planning and conduct of their activities in Antarctica", "to work together to better understand changes to the Antarctic climate and to implement actions consistent with the Paris Agreement's goals", and "to safeguard the Antarctic environment and dependent and associated ecosystems and to remain vigilant and continue to identify and effectively address current and future Antarctic environmental challenges by taking effective and timely action." Our work demonstrates that securing the future of Antarctic biodiversity will require the Antarctic Treaty Parties to implement regional conservation actions, while simultaneously working through relevant global frameworks to encourage all nations to deliver the aims of the Paris Climate Agreement.

## Materials and methods

Priority threat management (PTM) is a structured decision science approach that combines expert elicitation and scientific data to identify optimal and cost-efficient threat management strategies for conserving biodiversity across a region [36,37]. It has been successfully applied in regions around Australia [90,91], Canada [92], and Indonesia [93]. The method brings together relevant stakeholders in a workshop to define appropriate biodiversity features, management strategies, costs and feasibility, and utilises expert knowledge to derive predicted benefits to biodiversity for each management strategy. Further details of the PTM background and method are described in [37], and see definitions of PTM terms in Table J in S1 Text.

PTM is an ideal tool for the Antarctic as the remoteness and comparatively young nature of Antarctic science result in the region being data poor. This is especially prominent in the biological sciences, where remote areas of unsurveyed wilderness and severe logistic and funding constraints have resulted in a lack of quantitative data on comprehensive species taxonomy and distributions, and limited understanding of physiology and ecology. Where data and expert knowledge on biodiversity and potential conservation strategies do exist, they are often held in disparate sources, including the separation of expertise by subject matter. Thus, a process for bringing together expert knowledge is crucial for underpinning conservation plans and prioritisations alongside empirical data.

We followed the PTM approach, adapting it as required to suit the Antarctic biodiversity threat management problem. This consisted of defining the region, timeframe, and climate change scenarios pre-workshop, using expert elicitation during the workshop to define the threats, taxonomic groups, management strategies, costs and feasibility of each strategy, and to assess the benefit of each strategy for each selected taxon. Post-workshop, we quality checked the data and undertook cost-effectiveness, complementarity, and sensitivity analyses. Ethical clearance for this project was granted by the Commonwealth Scientific and Industrial Research Organisation (066/17).

### Region, timeframe, and climate scenarios

The Antarctic region was defined as the area subject to Antarctic Treaty governance, which includes all ocean, ice, and landmass south of 60˚S [94]. The end of the current century, i.e., 2100, was selected as the timeframe for the project, 83 years from the time of the workshop.

Two IPCC RCP climate forcing scenarios (RCP4.5, 8.5) were selected for the assessments and analysis [42]. The RCP4.5 pathway is based on moderate carbon emissions, and RCP8.5 represents more severe emissions, without climate mitigation policies [42]. While the RCPs were the current climate forcing scenarios in use at the time of this research by the IPCC, recent research has suggested that RCP8.5 represents an unlikely future scenario as emissions have fallen over the last decade [95], with the uncertainty around climate forcing scenarios highlighting the importance of comparing multiple scenarios. For comparison, the RCPs are succeeded by the Shared Socioeconomic Pathways (SSPs) in the latest IPCC AR6 synthesis, where RCP8.5 is replaced by SSP5-8.5, RCP4.5 by SSP2-4.5, and RCP2.6 by SSP1-2.6 [95,96]. We did not use a scenario representing "no climate change" as this was considered unrealistic [97]. However, one of the outcomes of the successful implementation of the "Influence external policy" strategy would be sufficient reductions in global carbon emissions to achieve the Paris Climate Agreement target of less than 2˚C by 2100 [38]. Therefore, RCP2.6, the low emissions scenario and the only one to keep global warming to <2˚C [42,97], was used to assess the "Influence external policy" and "All Strategies Combined" strategies, instead of RCP4.5 or 8.5.

We divided the assessment area into 2 regions—the Antarctic Peninsula (including the South Shetland and South Orkney archipelagos) and continental Antarctica (where biodiversity is primarily concentrated around the coasts). These regions are climatically [51,98] and biologically [99] distinct and represent a broad biogeographical division [1,100]. They also have largely (but not completely) distinct logistic and operational considerations [101].

### Expert participants and elicitation

The participants in the PTM process included 29 diverse experts (4 participated remotely), all of which were invited to be coauthors on this paper. We aimed to ensure that their expertise covered a variety of NAPs operating in different regions of the continent, with comprehensive knowledge in at least one of the following subjects: biodiversity, policymaking, logistics, tourism, or conservation. The experts were drawn from 12 different countries, with representatives from every continent. The group had balanced gender representation and included early career researchers. The expert elicitation was primarily carried out during a 2-day workshop held at Katholieke Universiteit Leuven in Belgium in July 2017, though extensive preparatory work and follow up was required. Participation involved structured sessions at the workshop, as well as pre-workshop and follow up communication: to enable the definition and parameterisation of biodiversity features, threats, timeframe, spatial units, scenarios, as well as the finalisation of the list of management strategies and their costs, feasibility and benefits (S1 Data). An independent facilitator with no Antarctic affiliation helped to run the workshop and ensured all the experts' contributions were heard and accounted for.

### Selection of taxonomic groups

We focused on Antarctic terrestrial biodiversity, which includes microbes, terrestrial algae, various invertebrate groups (nematodes, tardigrades, rotifers, enchytraeid worms, springtails, mites, and 2 insects), mosses and liverworts, lichens, and 2 vascular plants [99,102]. A number of marine seabirds, including 4 species of penguin, were also included as they rely on ice-free areas for breeding [103], or in the case of most colonies of the emperor penguin, fast ice [48].

The biodiversity experts (a subset of all experts) selected a total of 38 biodiversity taxonomic groups to include in the exercise (hereafter "taxon/taxa"), which ranged from single species to broader functional taxa (Table K in S1 Text). Each taxon contained species that were expected to respond similarly and by similar magnitudes to threatening processes and management strategies. The taxonomic resolution of groups differed depending on their predicted response and level of species-specific knowledge available. For example, most seabird species were considered as individual taxa, while soil microfauna were grouped based on habitat requirements (e.g., moss-associated tardigrades, rotifers and nematodes formed a single group containing multiple higher taxa). Seabirds in the procellariid taxa were grouped as experts felt they would respond in similar ways to threats, though Southern Giant Petrels were kept separate as there is more information available. A further 7 taxa identified were not included in the analyses as none of the participating experts felt sufficiently knowledgeable to assess them. These included: thalloid liverworts, snow algae, freshwater decapods and copepods, non-marine aquatic system plankton, and sea-ice dependent and sea-ice independent seals.

### Threats

Prior to the workshop, experts identified multiple threatening processes that could impact terrestrial Antarctic biodiversity by 2100, including direct and indirect impacts of climate change, human activity (science and tourism), non-native species, and pollution (Table L in S1 Text). Some marine-based threats, such as fishing and ocean acidification, were considered in the

study due to their potential direct (e.g., krill-dependent penguins impacted by fisheries) or indirect (e.g., redistribution of penguin colonies can impact distribution of vegetation via nutrient inputs) impacts on terrestrial ecosystems, including marine seabirds. Background information on, and predictions for, some of these threats on a 2100 timeframe were provided to the experts to assist with their decisions, including maps of projected changes in temperature, precipitation, ice-free areas, and non-native species for RCP4.5 and RCP 8.5 across the Antarctic continent [25,51].

## Management strategies

Experts agreed on a total of 13 strategies for managing and conserving Antarctic terrestrial biodiversity to 2100 (Table 1; S1 Data). A quantifiable aim was determined for each strategy (Table 1). While differing aims may result in a different prioritisation, a large amount of time was spent carefully defining these aims as this is paramount to ensuring useful outcomes, as it is in any conservation planning exercise [104]. Strategies had to be stand-alone, providing a substantial benefit to biodiversity, without being contingent on the implementation of another strategy to successfully meet their aims. If strategies were considered to be too interdependent, they were amalgamated (for example, managing individual species on the ground and protecting individual species via policy were combined to form the "Manage and protecting species" strategy). All of the strategies applied to both regions, except for "Protect vegetation from physical impacts" which only applied in the Antarctic Peninsula, where trampling of vegetation by seal species occurs.

The first strategy defined was "Business as usual," where conservation actions currently in place are continued, but no new actions are added. The regularity of implementing current conservation actions was not considered to change over the timeframe in the "Business as usual" strategy as current conservation actions, such as biosecurity or designating a new protected area, are already irregularly implemented and any change (increase) in the frequency of implementation would constitute one of the new management strategies. This strategy represented a baseline against which other strategies could be evaluated. The other strategies included managing infrastructure, transport, human activity, vegetation, non-native species, area protection, and protecting species. The "Influence external policy" strategy objective aimed to influence global policy to reduce pressures on Antarctica from external threats, i.e., primarily climate change, but also including other threats, such as microplastics and persistent organic pollutants (POPs). The final 2 strategies "All strategies combined excluding policy influence," and "All strategies combined" are a combination of the other strategies, used to assess whether the expected benefits from strategy combinations are greater than the sum of their parts. Because the "All strategies combined" strategy includes "Influence external policy," it was also assessed under the RCP2.6 climate forcing scenario. The combination strategy "All strategies combined excluding policy influence" was utilised in order to assess the benefits of all the other strategies if the "Influence external policy" strategy was not implemented, or not successful, and the world were committed to >2˚C warming (i.e., RCP4.5 or RCP8.5).

Each strategy consisted of a suite of actions determined by the experts which, when implemented together, ensure the strategy can successfully meet its objective/s. The actions are detailed in S1 Data.

## Feasibility and cost

Feasibility consisted of 2 aspects: "likelihood of uptake" and "likelihood of success," that were estimated by the experts using a scale of 0% to 100%, where: 0 = Impossible, 15 = Improbable, 25 = Unlikely, 50 = Fifty-fifty, 75 = Likely, 85 = Probable, 100 = Certain. Likelihood of uptake

was estimated for each action and represents the likelihood that policymakers will agree to implement this action considering social and political factors, assuming that cost is not a barrier. The likelihood of uptake estimates were averaged for all actions in the strategy to obtain an overall estimate per strategy, which will be lower overall for strategies that include 1 or more actions that are more sociopolitically challenging. Likelihood of success was estimated per strategy and represents the likelihood that the strategy will meet its objectives in reducing impacts of threats, assuming that all actions have been implemented. Overall feasibility per strategy was calculated as a product of the 2 likelihoods and is provided as a probability (Fig C in S1 Text and S1 Data).

Experts estimated the cost for each action over the 83-year time horizon being considered (the costs are outlined in detail in S1 Data). Where available, they used existing cost information to help inform their estimates. Costs typically consisted of: number of FTE employees, number of Antarctic berths required (researcher living costs and transport), and other costs (e.g., number of workshops, laboratory analyses). The cost of FTEs (in Antarctica and not in Antarctica), berths (short summer, long summer, and winter), and other costs that were used multiple times (such as workshops) were standardised across strategies using the average USD value of cost estimates given from multiple NAPs (Table M in S1 Text). Cost estimates used 30 operating NAPs, including 35 seasonal and 45 year-round operational stations. Costs were estimated assuming all actions are successfully implemented.

The total costs of the strategies over the whole timeframe were converted to present-day values (present value: PV) using a conservative social discount rate of 2%, where future costs are discounted to present-day values. Discounting costs is important to allow for fair comparison of strategies costs when the payment schedule of actions differs over time. For example, some strategies have large start-up costs or require substantial investment for only part of the timeframe (e.g., "Remediation" has a large start-up cost and ongoing investment for the first 50 years, before teetering off when the remediation is finished), while others have recurring costs distributed more regularly across the years (e.g., "Minimise impacts of human activity" has costs distributed much more evenly and recurring actions every 2, 3, and 5 years). PVs were also compared using discount rates of 0% and 5% (Fig Da in S1 Text). PVs were calculated prior to the COVID-19 pandemic and do not incorporate the likely impacts and expected economic recession [105], which are likely to impact both governmental and nongovernmental Antarctic operations well beyond the short term [18,106].

The magnitude of costs remained the same regardless of the discount rate applied, and it had little impact on the order of top-ranking cost-effective strategies (Fig Da in S1 Text). However, the management intensity and scale of the actions underlying each strategy, and hence their costs, may be considered subjective. In some cases, higher investments would achieve additional benefits. For example, in managing non-native species, a larger investment in biosecurity is likely to increase biodiversity benefits. Some management strategies also include large amounts of preparatory work to optimally make decisions and take appropriate actions. The primary example is baseline biodiversity surveys undertaken in the "Manage non-native species and disease" strategy and "Managing and protecting species" strategy, which substantially increase the cost of these strategies, and highlight that our current understanding is limited by uncertainty and data-deficiency. Also, this was an Antarctic PTM exercise, and the goal was to prioritise strategies and actions that can be undertaken in the Antarctic and where Antarctic policy can have an influence. Thus, while we identified a strategy to influence global external policy, the actions were targeted at the Antarctic stakeholders' contributions and did not include the global costs of reducing carbon emissions, such as increasing use of renewable energy sources, carbon sequestration, or negative emissions technologies [107,108]. However,

the strategies were consistently costed as the actions required to meet their aims, and the experts revised them several times to ensure this.

To simplify the process, and because the experts already had a laborious task, it was assumed that costs and feasibility would remain the same for both RCP4.5 and RCP8.5. Costs were adjusted for regional analysis in proportion to the regional strategy effort required, where experts estimated the percentage effort of each overall strategy that should be targeted towards each region (Table C in S1 Text).

## Benefit assessments

To assess the predicted benefits of each management strategy, we needed an appropriate metric. Often in PTM, and in other conservation assessments, this metric is probability of persistence or extinction risk, where the predicted benefit of a strategy is the estimated reduction in extinction risk to the species [36,37]. The experts determined that extinction risk is not relevant to most Antarctic terrestrial species on a 2100 timeframe as populations change slowly. Thus, we defined a more flexible metric for use in this study—"intactness"—that represents how intact or unharmed the taxonomic group is relative to a baseline. Experts were encouraged to conceptualise intactness in an appropriate way for each taxonomic group, for example, extinction risk, population decline, or functional persistence might be appropriate for vertebrates, but range contraction, ground cover, or density might be more appropriate for vegetation or invertebrates, which are more unlikely to become extinct by 2100.

To estimate the benefits of each management strategy, biodiversity experts individually predicted the intactness of each biodiversity taxon at 2100, first under the "Business as usual" baseline strategy and then again under each of the 12 strategies, assuming each strategy had been successfully implemented and would meet its objectives to reduce the impacts of threats. Intactness values were always estimated relative to the intactness of the taxon today (2017, at the time of the workshop), where a value of 100 indicates that the "intactness for the taxon is the same as today," a value of 0 represents that the "taxon is completely degraded relative to today," and a value of 200 represents that the "taxon is doing twice as well as today." The experts considered a scale of 0 to 200 necessary for Antarctic taxa, as some groups are predicted or already observed to benefit from climate warming (e.g., [1,4,50,109]). Such taxa expanding beyond current intactness may also have consequential negative impacts on other species, such as the grass *D. antarctica*'s predicted ability to outcompete native mosses due to more efficient nitrogen acquisition [110].

Intactness estimates were recorded by biodiversity experts using an online tool created with JavaScript and HTML and hosted by a NodeJS web server on Amazon Web Services. The tool effectively functioned as a survey where biodiversity experts could select the taxon they wished to assess and then proceed through the questions estimating the benefits for both regions (Antarctic Peninsula and continental Antarctica) and for both climate forcing scenarios (RCP4.5 and RCP8.5) using a slider. The experts provided a best guess, lower bound, and upper bound for each question, as is standard for structured expert elicitation [39]. The lower bound represented what the experts believe to be the lowest possible (pessimistic) intactness value for the taxon by 2100, the upper bound represented the highest possible (optimistic) intactness value, and the best guess represented their best estimate of the true value. They also provided a confidence value, which represented their confidence that the true value lay within the range of the lower to upper bound [81]. Experts were also able to enter comments and identify sources of uncertainty in their estimates, categorised as knowledge shortfalls.

Every effort was made to ensure each taxon was assessed by multiple biodiversity experts. Nevertheless, 6 taxa had only 1 assessor while some had as many as 7, as the biodiversity

experts only assessed taxa they felt they had sufficient expertise to undertake the assessment for. After the workshop, the biodiversity experts' intactness values were anonymised, and all estimates (for each taxon) were circulated to allow opportunity for revision of values in the light of the judgements of the expert group as a whole, as per the modified Delphi method [40,81]. Expert values were then averaged for use in the analysis.

Because some taxa were predicted to expand under climate change (i.e., increase intactness beyond the 100% values of today), we performed all analyses twice: first using expert values capped at 100% before averaging, so that the benefits of taxa expanding above 100% intactness were excluded from the totals (Analysis 1 –"An1"), and second including all benefits (Analysis 2 –"An2"). The primary results of this paper focus on An1 as by definition, conservation is concerned with abating biodiversity loss or impact, so species that increase intactness with climate change do not necessarily require conservation. Thus, we chose to focus analyses on those taxa predicted to decline (decrease intactness) by the end of 2100 (though results also including benefits for those taxa for which climate change may have a positive impact are provided in Fig A and Tables H, I, and N in S1 Text).

Expert's values of predicted future baseline intactness allowed us to identify which taxa are likely to be vulnerable to future changes in the region. Some taxa that are predicted to have future intactness values similar to current (100%) levels are classified as vulnerable in Analysis 1, but as benefiting in Analysis 2. This occurs when the averaged expert intactness value is below 100% when the values are capped, but where the average might be above 100% when uncapped (see Table N in S1 Text, and raw values on the Australian Antarctic Data Centre; https://doi.org/10.26179/5da8f8e7a2256). These results highlight some of the uncertainty in predicting future responses to global change of some taxa, though we consider identifying them as potentially vulnerable to be appropriately conservative from a conservation perspective.

Some experts predicted that the 2 vascular plants, *C. quitensis* and *D. antarctica*, may expand into the continental region of Antarctica (from the Peninsula region) as climate changes. These benefits were included for the Analysis 2 benefits calculations, though were excluded from vulnerability assessments.

## Knowledge shortfalls

Seven knowledge shortfalls were identified by [41] that represent gaps in our understanding of biodiversity data and hinder our abilities to answer hypotheses or make decisions, such as a limited understanding of species distributions (defined as a "Wallacean" shortfall). An 8th shortfall was identified prior to the workshop by Peter Fretwell (British Antarctic Survey) and colleagues, representing a technological shortfall (termed a "Galilean" shortfall) where the data are available, but where we do not yet possess the skills or computing power to understand or interpret them adequately. The shortfalls are outlined in Table A in S1 Text. The experts identified relevant shortfalls for each biodiversity taxon in an effort to better understand the drivers of uncertainty in benefit estimates (Table B in S1 Text). The shortfalls also provide useful information for directing future research to reduce data gaps and uncertainty.

## Total potential benefits

The potential benefit per strategy was calculated as the difference between the best estimate intactness value of the strategy and the intactness value of the baseline (error bars were generated giving the difference between the upper/lower bound intactness values and the baseline values). The benefit calculations thus use the arithmetic, or absolute, difference to calculate benefit. An alternative method would be to use percentage change, which would class increases

in intactness for those with a lower baseline (e.g., intactness baseline value of 5% increasing to 30% with strategy, percentage change of 500%) as more significant than those with a higher baseline (e.g., intactness baseline value of 70% increasing to 95% with strategy, percentage change of 35.7%), whereas this increase (25%) is the same using arithmetic difference. Percentage change could be useful if prioritising individual taxa for management as it gives more weight to vulnerable taxa. We used arithmetic difference in this analysis as we focused on conservation outcomes overall and did not want to weight vulnerable taxa as more important.

It should be noted that, because the amount of benefit from implementing a strategy does not have to be the same for all 3 bounds, the benefits calculated for the upper and lower bounds can end up being both higher or lower than the benefit calculated for the best guess estimate. The potential benefit for "Influencing external policy" and "All strategies combined" was calculated in the same way as for all the other strategies, noting the assumption that the future with the strategy in place (an outcome of the successful strategy implementation) was a reduction in climate forcing scenario to RCP2.6 relative to the RCP4.5 or RCP8.5 baseline.

Benefits were summarised at different scales, including potential benefits per taxon, per region, and per climate scenario, and then summed to give regional and overall potential benefits for all biodiversity taxa combined. We assume that benefits accrue over time at a similar rate up to the maximum improvement estimated for each action. Hence for ease of interpretation, we do not discount the benefit metric over time as we do for costs.

## Cost-effectiveness

The cost-effectiveness of each strategy $i$ ($CE_i$) was calculated as the total potential benefit (sum benefits across all taxa) ($B_i$) divided by the expected cost ($C_i$) and multiplied by feasibility ($F_i$):

$$CE_i = \frac{B_i F_i}{C_i}.$$

The CE score provides an indication of the total expected benefit that is likely to be achieved per unit cost spent, for each strategy that decision makers choose to attempt to implement. Strategies were ranked according to cost-effectiveness score and rankings compared across regions, climate scenarios, and taxa.

The primary results of the cost-effectiveness analysis presented in the main text use the best guess estimates provided by the experts, though results for the upper and lower bounds are available in Table E in S1 Text. Because the "Influence external policy" strategy includes only the Antarctic component of working toward climate mitigation, and not the full global cost of reducing emissions, it was excluded from the primary cost-effectiveness and complementarity analyses.

## Complementarity analysis

While cost-effectiveness analyses are a convenient tool to rank strategies independently, complementarity analyses are more suitable for identifying and prioritising subsets of strategies to guide conservation investment under different budgets. A complementarity analysis ensures a maximum coverage of biodiversity benefits by identifying sets of strategies that complement each other in order to achieve equity of benefits across taxa and avoid redundancy (i.e., one taxon receiving maximum benefits and another taxon receiving none). This is achieved by solving a combinatorial optimisation problem also termed a complementarity analysis [37,111].

Complementarity analyses rely on setting appropriate persistence (intactness) thresholds for biodiversity, to ensure as many taxa as possible reach the designated threshold. We used

70%, 80%, and 90% of expected intactness as our designated thresholds, where a value of 100% represents the taxon's current intactness. We considered these values appropriate for Antarctic biodiversity, which does not face the same levels of extinction and rapid human impacts as global biodiversity, and for which any decline in intactness is considered substantial.

We used the best guess estimates of biodiversity intactness provided by the experts, capped at 100% to exclude benefits to those taxa expected to benefit from climate change, and a 2% social discount rate on PV to undertake the complementarity analysis. The expected intactness $P_{ij}$ for each taxon $j$ under a given strategy $i$ can be calculated as follows:

$$P_{ij} = \bar{P}_{baselinej} + F_i \bar{B}_{ij},$$

with $\bar{P}_{baselinej}$ the mean intactness across contributing experts assuming the baseline strategy applies to taxon $j$, $F_i$ the feasibility of strategy $i$, and $\bar{B}_{ij}$ the mean potential benefit across contributing experts of applying strategy $i$ for taxon $j$. Conveniently, the $P_{ij}$ can be assessed under different persistence thresholds and provide a binary matrix T such that an element $T_{ij}$ takes value 1 if the expected persistence of taxon $j$ under strategy $i$ ($P_{ij}$) is above the designated threshold and 0 otherwise. Note that the baseline represents the future intactness of a taxon subject to all threats (and only current conservation actions), thus inaction in addressing one threat may be negatively affecting intactness, while a strategy that addresses a different threat may be simultaneously positively affecting intactness. Therefore, a strategy must increase intactness of a taxon enough to reach the persistence threshold while potentially still being negatively affected by other threats.

The complementarity problem is usually formulated as maximising the number of taxa above the designated threshold for a given budget [37]. Here, we chose to minimise the cost of securing a given number of taxa above threshold. Both problems are equally difficult to solve —classified as NP-hard in the complexity scale for decision problems [112]. In our case, the aim was to generate a complete investment profile to secure as many taxa as possible. Because we had a finite number of taxa, it was computationally advantageous to formulate the complementarity problem as a variant of a minimum set coverage problem, i.e., minimise the cost of securing a given number of taxa above threshold [113]. This combinatorial problem can be formulated as an integer linear program (ILP; [114,115]). Here, we define $S$ as the finite set of strategies and $R$ the finite set of taxa. Formally, we sought to minimise the cost of selected strategies:

|  | $\text{minimise}_{y_i} \sum_{i \in S} y_i C_i$ |  |
|---|---|---|
| Subject to |  |  |
| $\forall i \in S, \forall j \in R,$ | $x_{ij} \leq y_i$ | (c1) |
| $\forall j \in R,$ | $\sum_{i \in S} x_{ij} \leq 1$ | (c2) |
|  | $\sum_{i \in S} \sum_{j \in R} x_{ij} T_{ij} \geq Target$ | (c3) |
|  | $y_i \in \{0,1\}, x_{ij} \in \{0,1\}, i \in S, j \in R$ | (c4) |

Solving this ILP requires finding the values of 2 sets of binary decision variables. The first set of decision variables $y_i$ determines if a strategy $i \in S$ is selected ($y_i = 1$) or not ($y_i = 0$). If strategy $i$ is selected, the taxon $j$ with values of $T_{ij} = 1$ is assumed secured. The second set of auxiliary decision variables $x_{ij}$ identifies the taxon $j$ secured by strategy $i$—these decision variables ensure we reach the target of a given number of taxa to secure (identified by Target). We sought to minimise the cost of implementing strategies while respecting 4 constraints:

- (c1): A strategy is selected ($y_i = 1$), if at least 1 taxon is secured by strategy $i$.

- (c2): For the purpose of the optimisation, only 1 strategy can count towards securing a taxon above the threshold. Therefore, the sum of the $\sum_j x_{ij}$ represents the total number of taxa secured.

- (c3): The number of taxa secured must be greater or equal to the given target number (Target).

- (c4): The decision variables are defined as binary.

Running this ILP using Target values ranging from 1 taxon to 38 taxa and 2 RCP scenarios, generated Fig 3.

As only 1 strategy can count toward securing a taxon above a threshold, strategies are considered independently of one another (except for "All strategies combined excl. IEP") and interactions between strategies are not considered (i.e., if multiple strategies provide benefits to a taxon, the benefits are not considered cumulative). If 2 or more strategies help to secure the same taxa above the threshold, then the one that is cheaper is selected (because selecting more than one would be redundant as those particular taxa can reach the threshold under either strategy). If multiple strategies help to secure different taxa above the threshold, then they are complementary (because they increase the number of taxa that can reach the threshold). If 2 or more strategies help to secure some of the same and some different taxa, then while there may be some redundancy of benefits for those individual same taxa (because they can reach the threshold under either strategy), overall, for all taxa, there is no redundancy as both strategies are still required to secure the different taxa.

In reality, there may be some interactions between strategies, and benefits might be cumulative or redundant when multiple strategies are implemented together. For example, a taxon may not be able to reach a threshold if either strategy *a* or strategy *b* were implemented but might be able to reach the threshold if strategy *a* and *b* are implemented together because the benefits are cumulative. Accounting for these factors in complementarity analysis would require the experts to estimate the expected benefits of not just each individual strategy, but every possible combination of strategies, while considering whether there was redundancy or not between each combination. We deemed this too complex and too laborious a task and instead utilised the combined strategies "All strategies combined" and "All strategies combined excl. IEP" to provide estimates of the maximum total benefits that can be achieved if all strategies were to be implemented.

## Sensitivity analyses

We performed sensitivity analyses to examine the impact of uncertainty on the cost-effectiveness rankings. The experts provided 4 values for each intactness estimate they made: an upper bound, a lower bound, a best estimate, and a confidence estimate representing their confidence that the true value lay within the range of the lower to upper bound. Because the experts did not provide the same confidence estimates, we standardised the intactness estimates using linear extrapolation of the lower, best, and upper bounds, and confidence interval to fit 80% credible bounds (confidence) around individuals' best estimates (as per [40,116]). The absolute minimum and maximum bounds computed from the linear extrapolation (representing the 10th and 90th percentile, respectively), together with the best estimates, were used to constrain beta-pert distributions for each expert's estimates per taxa. Monte Carlo simulations with 10,000 iterations, seeded with a random number, sampled values from the constrained beta-pert distribution for each taxon (as per [37,82]), which were used to examine uncertainty of

the intactness estimates (converted to benefits) when incorporated into cost-effectiveness analysis (Fig Ea in S1 Text; using the original estimates of cost and feasibility). To avoid experts' fatigue, lower or upper bounds for the cost or feasibility estimates were not elicited. Instead, we used 70% and 130% of the original values [91] to produce uniform distributions of cost and feasibility for each strategy, from which the 10,000 samples were drawn for examining uncertainty of cost and feasibility (Fig Eb and Ec in S1 Text). Finally, we combined the 10,000 samples of intactness, cost, and feasibility to examine cost-effectiveness robustness in light of all 3 components of uncertainty combined (Fig F in S1 Text).

Additional information: Antarctic PTM database containing intactness values, benefits, and uncertainties for each biodiversity taxon has been made available through the Australian Antarctic Data Centre (AADC; https://doi.org/10.26179/5da8f8e7a2256).

## Supporting information

**S1 Data. Table of proposed management strategies for conserving Antarctic biodiversity to the end of this century, detailing strategy objectives, specific actions that make up the strategy, costs, and feasibility of each action.**
(XLSX)

**S2 Data. Workbook containing the numerical values underlying the figures in the main text and Supporting information.** The values for each figure are provided in separate worksheets. These values were calculated from the raw values using the methods described in the "Materials and methods" section of the main text. The cost and feasibility values are available in S1 Data. The averaged expert intactness and benefit values are available on the Australian Antarctic Data Centre: https://doi.org/10.26179/5da8f8e7a2256.
(XLSX)

**S1 Text. PDF file containing the supporting figures and tables for "Threat management priorities for conserving Antarctic biodiversity".** File contains Figs A–F and Tables A–N.
(PDF)

## Acknowledgments

We thank the organisers of the SCAR Biology 2017 Symposium at Katholieke Universiteit Leuven in Belgium, especially Anton Van de Putte and Bruno Danis, for organising workshop rooms and schedules. We thank Patricia Ortúzar for her input and participation in the 2017 workshop, Craig Salt who helped to facilitate the workshop, and Jenny Baeseman who provided administrative support. We thank A. Lynnes, A. Hobday, and H. Murphy for valuable feedback on earlier drafts. D.H.W. acknowledges support of NSF MCM LTER. Publication of this research was supported by the Integrated Digital East Antarctica Program. It is a contribution to the Integrated Science to Inform Antarctic and Southern Ocean Conservation (Ant-ICON) Scientific Research Programme of the Scientific Committee on Antarctic Research (SCAR).

## Author Contributions

**Conceptualization:** Iadine Chadès.

**Data curation:** Jasmine R. Lee.

**Formal analysis:** Jasmine R. Lee, Aleks Terauds, Josie Carwardine, Justine D. Shaw, Richard A. Fuller, Hugh P. Possingham, Iadine Chadès.

**Funding acquisition:** Jasmine R. Lee, Aleks Terauds.

**Investigation:** Jasmine R. Lee, Aleks Terauds, Josie Carwardine, Justine D. Shaw,
Steven L. Chown, Peter Convey, Neil Gilbert, Kevin A. Hughes, Ewan McIvor,
Sharon A. Robinson, Yan Ropert-Coudert, Dana M. Bergstrom, Elisabeth M. Biersma,
Claire Christian, Don A. Cowan, Yves Frenot, Stéphanie Jenouvrier, Lisa Kelley,
Heather J. Lynch, Birgit Njåstad, Antonio Quesada, Ricardo M. Roura, E. Ashley Shaw,
Damon Stanwell-Smith, Megumu Tsujimoto, Diana H. Wall, Annick Wilmotte,
Iadine Chadès.

**Methodology:** Jasmine R. Lee, Aleks Terauds, Josie Carwardine, Justine D. Shaw,
Richard A. Fuller, Hugh P. Possingham, Iadine Chadès.

**Project administration:** Jasmine R. Lee, Aleks Terauds, Josie Carwardine, Iadine Chadès.

**Resources:** Jasmine R. Lee.

**Software:** Jasmine R. Lee, Michael J. Lee, Iadine Chadès.

**Supervision:** Aleks Terauds, Justine D. Shaw, Richard A. Fuller, Hugh P. Possingham,
Iadine Chadès.

**Validation:** Jasmine R. Lee, Josie Carwardine, Iadine Chadès.

**Visualization:** Jasmine R. Lee, Aleks Terauds, Josie Carwardine, Justine D. Shaw,
Richard A. Fuller, Hugh P. Possingham, Iadine Chadès.

**Writing – original draft:** Jasmine R. Lee, Aleks Terauds, Josie Carwardine, Justine D. Shaw,
Richard A. Fuller, Hugh P. Possingham, Steven L. Chown, Peter Convey, Kevin A. Hughes,
Ewan McIvor, Heather J. Lynch, Iadine Chadès.

**Writing – review & editing:** Jasmine R. Lee, Aleks Terauds, Josie Carwardine,
Justine D. Shaw, Richard A. Fuller, Hugh P. Possingham, Steven L. Chown, Peter Convey,
Neil Gilbert, Kevin A. Hughes, Ewan McIvor, Sharon A. Robinson, Yan Ropert-Coudert,
Dana M. Bergstrom, Elisabeth M. Biersma, Claire Christian, Don A. Cowan, Yves Frenot,
Stéphanie Jenouvrier, Lisa Kelley, Michael J. Lee, Heather J. Lynch, Birgit Njåstad,
Antonio Quesada, Ricardo M. Roura, E. Ashley Shaw, Damon Stanwell-Smith,
Megumu Tsujimoto, Diana H. Wall, Annick Wilmotte, Iadine Chadès.

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
