## [Editor Report · Decision Letter 0]

13 Sep 2021

Dear Dr Lee, 

Thank you for submitting your manuscript entitled "Threat management priorities for conserving Antarctic biodiversity" for consideration as a Research Article by PLOS Biology.

Your manuscript has now been evaluated by the PLOS Biology editorial staff, as well as by an academic editor with relevant expertise, and I'm writing to let you know that we would like to send your submission out for external peer review.

Please re-submit your manuscript within two working days, i.e. by Sep 15 2021 11:59PM.

Kind regards,

Roli Roberts

Roland Roberts

Senior Editor

PLOS Biology

rroberts@plos.org

---

## [Decision Letter · Decision Letter 1]

25 Nov 2021

Dear Dr Lee,

Thank you for submitting your manuscript "Threat management priorities for conserving Antarctic biodiversity" for consideration as a Research Article at PLOS Biology. Your manuscript has been evaluated by the PLOS Biology editors, an Academic Editor with relevant expertise, and by three independent reviewers.

IMPORTANT: You’ll see that there is a strong split in opinion among the reviewers. Reviewer #3 is very positive, and reviewer #1 somewhat so. However, rev #1 wants you to re-structure the paper and give substantially more methodological detail. Reviewer #2 recommended that we Reject your paper, raising three major concerns. After some discussion with the Academic Editor, we'd like to give you an opportunity to review, but I shall transmit the following [somewhat edited] comments from the Academic Editor to give you an idea of how we think you should address this reviewer's concerns: "Reviewer #2 also seems to echo my original concerns by accepting that expert guesses are necessary but asking how good are these "guesses"?... One could ask how often experts actually get it right?... I am trying to be somewhat provocative, but this question from reviewer #2 does seem like the linchpin of any attempt to revise the paper... I think the paper will ultimately be substantially improved by challenging the authors to address the comments from #2 with actual revisions to the text and analyses."

In light of the reviews (below), we will not be able to accept the current version of the manuscript, but we would welcome re-submission of a much-revised version that takes into account the reviewers' comments. We cannot make any decision about publication until we have seen the revised manuscript and your response to the reviewers' comments. Your revised manuscript is also likely to be sent for further evaluation by the reviewers.

We expect to receive your revised manuscript within 3 months. 

**IMPORTANT - SUBMITTING YOUR REVISION**

*Re-submission Checklist*

*Published Peer Review*

*PLOS Data Policy*

Sincerely,

Roli Roberts

Roland Roberts

Senior Editor

PLOS Biology

rroberts@plos.org

REVIEWERS' COMMENTS:

Reviewer #1:

The conservation of Antarctic biodiversity is clearly of international interest and a subject that needs more to be written about it. I applaud the basic work done and the intention behind it. However, I am afraid that the manuscript, as written, is not yet of sufficient clarity to allow an adequate review. I have attempted to make a few comments along the way that might assist, and some of them will simply show the level of confusion engendered in the reader at present. But in general, can I suggest that the authors engage an experienced communicator to give more shape to what could clearly be an important study?

Introduction: Conservation ROI is controversial because of opposition to the idea "cheapest is best". At present, the introduction limits itself to posing the ROI question without set-up motivation for why it's worth asking. Given the controversy, I believe it's important for the authors to set up the motivation and acknowledge why the two questions being asked are important, and are different in their interest. 

Line 103: The authors write: "We apply a structured, participatory decision-science approach 104 (Carwardine et al. 2012, 2019) to quantify and prioritise cost-effective and complementary 105 strategies for achieving biodiversity conservation in terrestrial Antarctica (see Materials and 106 Methods). This approach is especially suited to the Antarctic as the continent's isolation, 107 extreme climate and the relatively late initiation of scientific activities (which accelerated 108 following the International Geophysical Year of 1957-1958; Summerhayes 2008) have resulted 109 in a lack of quantitative data, especially on species distributions, interactions and taxonomy". 

 This phrase suggests that the method was chosen a priori as some sort of ideal. There is a little bit of overenthusiastic marketing going on, I feel. There is nothing wrong with saying that the continent is data-poor and so one has to rely on expert-based approaches, acknowledging very briefly the strengths and limitations of such approaches. Line 110 can then be a context-specific defence of the approach for this study. As the sentence stands, it unfortunately implies special pleading, and many readers will actually lose confidence in what is to follow. It's the rhetorical difference between stating your approach and honestly defending it (with acknowledgment of any limitations), versus claiming that you chose the cutting-edge method as a claim to credibility. 

 Additionally, please start a new section that begins with whatever replaces this sentence. As it stands, it is not clear that the summary-of-methods has begun at this point. It would be much clearer if it were subtitled or otherwise better signposted.

Line 115: I suggest this also needs to be signposted - "for our survey" would do - and flow from the defence of expert-based methods. Also, the very short paragraph feels chopped up. 

Line 120: if the methods are based on benefit and cost, it feels incomplete to summarize only the benefit method. A sentence here about the cost approach would be best advised.

Line 130: this entire paragraph is confusing because it starts by saying there are two RCPs (4.5 and 8.5) and then immediately talks about RCP2.6 instead. Do the authors mean to say that they assessed the option of influencing climate policy as one threat abatement measure in isolation, and then asked the experts about the likely outcomes from a basket of non-policy-based, more local abatement measures under two different climate scenarios?

Line 148: please signpost with something like "In our results, we found that" to make it clearer what is happening here. Also, if there is going to be a results section on threat, then surely it is important to set up a study of threat in the questions asked and in the brief methods description? An alternative - and perhaps a better one - is to move this entire section into the online material. As it stands, it is a sudden rush of highly detailed, descriptive work that seems to confuse the reader about where the ms is going. Case studies are useful but when the paper is a high-level piece, it becomes difficult to follow if suddenly sidetracked into a long summary of case-specific details. 

Line 179: These details might be better off in a caveats section that is either in the online material, or more towards the end of the ms. It would help the reader to have a quicker transition from question->method->result->discussion, implication and caveat.

Lines 157-9: again, this seems to mix up sections of the logic flow. I would suggest putting the threat to Emporers as an important point of interest for the lay reader, near the top of the introduction, and then the assumed outcome of climate change mitigation on Emporers in a later section where the "influence policy" option is being set up/motivated. 

Line 194 and figure 1: Please define peninsula/continent and explain its importance before mentioning it here. The figure is important but highly confusing: it is not immediately apparent that part (a) continues down the page. The message that some taxa will do worse and some will do better is a major result but has got lost graphically, and was not clearly and simply presented in the text either. It would help greatly if these points were clarified, please. Also, there seems to be a graphical mix-up between response to all threats and response to climate change alone. I'm afraid it's entirely confusing and leaves the reader wondering what results are being presented at a fundamental level. Part (b) is similarly lacking in a legend or explanation: is this a spatially explicit mapping of threat levels and if so, where is the key? And in the caption, how are the expected changes in outcome from different levels of intervention being included and measured? This seems to be suddenly implied as a method with no explanation leading up to it.

Line 222: I am again confused. Firstly, are the authors really saying that climate change is by far the biggest threat, but achieving the Paris commitments will make the same difference whether we assume large or small forcing? That is extremely counterintuitive and demands explanation and defending (including a clearer sense of the timelines involved in threat abatement and species decline). More generally, I find it very hard to decipher what this paragraph is trying to present. 

Line 232: what is the meaning of the phrase "Whilst most strategies provide only benefits to taxa"? I assume the intention is to say that most interventions are expected to reduce decline but some could actually have perverse outcomes and increase decline for some taxa?

Line 302: the authors now state what was already a red flag - that it would be almost impossible to put a cost on "influencing global climate policy". It would also be nearly impossible to measure the outcome per dollar. It is therefore surprising that the authors only acknowledge that problem at this late stage and omit policy influencing from ROI assessment, and also that they see no difficulty in comparing the costs (including those for policy influencing) at other points in the ms.

Line 333: the key figure here is surely that the $1.9 billion is spread over most of the century and equates to $23 million per year. This is not what the average reader is going to take away from the abstract - including a policymaker - when the abstract only mentions the $1.9 billion. Indeed, it makes little sense to state a total budget that is expected to magically end in exactly 83 years time. Why not lead on the annual cost, which is far more compelling? Similarly, in lines 361-374, annualizing all values would make much more policy sense.

Reviewer #2:

The paper describes the results of a solicitation of expert opinion to assess priorities for threat management to conserve Antarctic biodiversity. The paper is written clearly and well-executed. The authors provide a very clear statement of the questions to be addressed (lines 97-98). I think the authors have done about as well as could be done given the limited set of information they have to work with. The editorial question is whether this is enough to justify publication in PLOS Biology. 

The main weakness of the paper is the lack of underlying data to support the kind of cost-effectiveness exercise the authors want to carry out. We just don't know that much about Antarctic biodiversity. There is very little hard evidence on which to base an assessment. How would various taxonomic groups respond to various conservation strategies (what are the benefits)? What are the costs of various strategies? What is the likely feasibility and the likelihood of success? The lack of hard evidence means we have to rely on expert judgement. The experts probably have as good of a guess as anyone about these questions, but how good these guesses are, is itself a guess. 

A second weakness, which again is largely outside the control of the authors, is that the main threat to Antarctic biodiversity seems to be climate change, which means that reducing threats lies well beyond the Antarctic. Climate change clearly poses a threat to some species. However, if it is really the case that "Some taxa do not appear to be sensitive to climate change and their intactness is expected to remain similar to current levels" (lines 168-169), then this makes it seem as if climate change it the overriding threat and little else matters, though later the authors indicate that other strategies benefit some taxa. On climate change, there is little that the authors of this paper can add to the large literature on costs and feasibility of various climate policies and pathways and I'm not convinced that they should devote space to talking about the larger costs or benefits of addressing climate change. 

A third weakness, but one on which the authors could do something more to address, is to convince those outside the Antarctic research community that conservationists should be paying greater attention to Antarctica. Because Antarctica is relatively species poor and "is relatively free of many of the environmental threats that beset the rest of the world" (lines 76-77), Antarctic conservation doesn't seem likely to rank highly in terms of global conservation priorities. If this is incorrect, the authors should provide some evidence or reasoning why Antarctic conservation should be a higher priority. I can think of potential grounds for making this case: it is not very costly to provide protection, and species in Antarctica are phylogenetically more distinct or have properties that make them scientifically interesting (e.g., the ability to live in a cold and hostile environment). The authors may have more compelling lines of argument. Because Antarctica is not on the radar screens of most conservation organizations (e.g., it was largely ignored in the Aichi Targets) and does not get the attention that coral reefs, mangroves, or rainforests receive, the authors should make the case for paying attention to Antarctica in the introduction. 

Given that benefits, feasibility, and costs are all uncertain, the resulting conservation return on investment that combines these elements in a ratio is likely to highly uncertain. I know it is a cheap comment by a referee and often hard for authors to actually do, but this is a case where doing sensitivity analysis given some distribution on benefits, feasibility, and cost, is really needed. Without some further analysis along these lines it is hard to know how much confidence to have in the rankings of various strategies. 

It is not clear to me what the conservation strategies are specifically. The descriptions in Table 1 are rather general. For example, what is involved in "Reduce and minimise impacts of existing infrastructure compared to current levels"? What does this actually entail? How much reduction is possible? What kinds of changes in ecosystems would results? What impacts would this likely have on biodiversity? How much would it cost? Even for an expert, it seems like it would be hard to answer questions about benefits, costs, and feasibility, based on such a general description of a strategy. If so, what should we really make of the return on investment numbers?

Reviewer #3:

[see attachment for fully formatted version]

This manuscript seeks to perform a large-scale assessment of threat management to biodiversity in Antarctica. The researchers employed a thorough expert solicitation process to do this work, with important results showcasing the most vulnerable taxa and also highlighting management paths forward. This research is timely and important; it fills a major knowledge gap while also clearly pointing to the urgent need for stronger management (and provides the useful information on the potential cost of such management actions). I thank the authors for their excellent work and look forward to seeing this manuscript in print! 

The manuscript is well written and clearly laid out with nice figures which help to illustrate the findings. Below I provide a few relatively minor suggested edits and comments.

Page 6, Line 106-109. Do we have a lack of taxonomy data or is it more accurately that we have a lack of baseline data (including abundance, distribution, etc.) regarding species?

Page 6, line 120-123 (and corresponding text in the Materials and Methods section further down). It was not clear if the exercise (and metric) for intactness was something that the authored devised or if this metric had been used before by others. It would be helpful to clarify (e.g., is the authors approach for measuring taxon intactness something they drew on from other studies? Is this an established approach?)

Figure 1. Nice figure. My only question/comment is: Why is there a category for “Procellariids” and a separate category for “Southern Giant Petrels”. Are the Giant petrels also considered in the Procellariid category or were they separated? (and why, it would be useful to clarify somewhere)

Figure 3 was not super easy to read. Any shorthand used in the figure should be explained in the caption (e.g., all stras excl IEP). On the figure, it would be useful to clearly label figure A as RCP 8.5 and B as RCP 4.5. 

Throughout the text, but especially in the areas which addresses penguins and flying seabirds, there was not mention of marine-based threats such as fishing. I realize marine threats were outside the scope of the paper, but it would be useful to note this limitation (perhaps in the discussion). Particularly around the Antarctic Peninsula where krill fishing is high, there is likely direct or indirect effects on krill dependent penguins and perhaps on some flying seabirds. 

Reference to RCP 8.5 throughout the text and especially Page 27, line 506. It was a sound approach that 4.5 and 8.5 were employed (given the recommendations towards using these pathways at the time the research was conducted). However, some more recent work has also suggested the errors with RCP 8.5 (how it is economically unfeasible) and suggested that scientists do not use 8.5. See e.g., the recent work by Hausfather: 

Hausfather, Z., & Peters, G. P. (2020a). Emissions–the ‘business as usual’ story is misleading. Nature 577, 618-620.

Hausfather, Z., & Peters, G. P. (2020b). RCP8. 5 is a problematic scenario for near-term emissions. Proceedings of the National Academy of Sciences, 117(45), 27791-27792.

I would suggest an edit to line 506 (perhaps considering the debate on what scenarios to use) or point to the most recent IPCC report and their rework of the pathways (to SSP). That said, having the consideration of RCP 2.6 is excellent and provides even more information and insight. 

One overall comment is that it would be useful to have the Main text clearly labeled between Introduction, Results, Discussion and Conclusion. 

The Conclusion is great, especially tying it to the recent declaration.

---

## [Decision Letter · Decision Letter 2]

8 Jun 2022

Dear Dr Lee,

Thank you for your patience while we considered your revised manuscript "Threat management priorities for conserving Antarctic biodiversity" for consideration as a Research Article at PLOS Biology. Your revised study has now been evaluated by the PLOS Biology editors, the Academic Editor and two of the original reviewers.

In light of the reviews, which you will find at the end of this email, we are pleased to offer you the opportunity to address the remaining points from the reviewers in a revision that we anticipate should not take you very long. We will then assess your revised manuscript and your response to the reviewers' comments with our Academic Editor aiming to avoid further rounds of peer-review, although might need to consult with the reviewers, depending on the nature of the revisions.

IMPORTANT: Please note that reviewer #1 has provided extensive comments in the attached file. Make sure that you address these points as well as the remaining requests from reviewer #2.

**IMPORTANT - SUBMITTING YOUR REVISION**

*Resubmission Checklist*

*Published Peer Review*

*PLOS Data Policy*

*Blot and Gel Data Policy*

Sincerely,

Roli Roberts

Roland Roberts, PhD

Senior Editor

PLOS Biology

rroberts@plos.org

REVIEWERS' COMMENTS:

Reviewer #1: 

IMPORTANT: Please see formatted comments in attached file!

Reviewer #2: I think the authors have done a good job of responding to my comments. I continue to think that the paper is well written and well executed and is probably the strongest paper that could be written on this subject. Given that the editorial decision is that this topic is of high interest, I think the paper should be published. 

The introduction does a much better job of motivating the importance of conserving Antarctic biodiversity. The treatment of climate change has been clarified and improved. I agree with the authors that climate change is the primary threat to Antarctic biodiversity and deserves to be fully discussed. I appreciate, however, that the authors do not include climate change in the discussion of cost-effective strategies, where costs refer more directly to actions undertaken by the Antarctic community. 

My main concern on the previous was about the weakness of the underlying information base, which is an unavoidable issue. The authors have undertaken a sensitivity analysis to show that the cost-effectiveness rankings "were reasonably robust to all sources of uncertainty." I couldn't find figures S5 and S6 in my document, so I was not able to verify whether I agree that the rankings are reasonably robust or not. Providing a sentence or two saying what changes in the ranking under what circumstances would help. I think their main point that the top choices remain top choices for a wide range out of outcomes is the right result to emphasize. The description of the sensitivity analysis starting on line 961 was somewhat opaque (at least to me). I did not understand what the authors meant when they said "The intactness estimates given by the experts were standardised using linear extrapolation of the lower, best, and upper bounds, and confidence interval to fit 80% credible bounds around individuals' best estimates." I think the authors have done a credible sensitivity analysis, but I would ask that they provide a bit clearer explanation of what they did. 

One last small point that perhaps could be worked into the paper is about experts and uncertainty. It has been found in the literature that experts often are over-confident and place too small uncertainty bounds around their estimates. This would inject a further note of caution into interpreting the results.

---

## [Editor Report · Decision Letter 3]

7 Nov 2022

Dear Jasmine,

Thank you for your patience while we considered your revised manuscript "Threat management priorities for conserving Antarctic biodiversity" for publication as a Research Article at PLOS Biology. This revised version of your manuscript has been evaluated by the PLOS Biology editors and the Academic Editor. Please accept my further profuse apologies for the extraordinary delay incurred here after a series of unusual circumstances.

Based on our Academic Editor's assessment of your revision, we are likely to accept this manuscript for publication, provided you satisfactorily address the remaining points raised by the Academic Editor and the following data and other policy-related requests.

IMPORTANT: Please address the following:

a) The Academic Editor says "I appreciate the intention to not assume whether managers might be interested more in percentage rather than arithmetic differences in intactness (pg 5 of the response). However, the authors are still - albeit implicitly - making an assumption about what managers might be interested in. One sentence acknowledging the difference between interpretations based on arithmetic vs percentage differences could help round off this issue."

b) The Academic Editor has a second request: "The authors acknowledge the concern about the regularity of implementation by 2100 might change (pg 7 of review), but it is unclear how this issue has been addressed in the text. Again, can there be one sentence identifying this issue and briefly explaining why it might/might not influence the interpretation of the results? I wonder if this goes into the "Uncertainty" section and maybe it's already sort of touched on at line 540."

c) Please hyphenate "threat-management" in the Title (as you have in the Abstract).

d) Please address my Data Policy requests below; specifically, we need you to supply the numerical values underlying Figs 1AB, 2AB ,3AB, 4, S1AB, S2, S3ABC, S4ABC, S5ABC, S6, either as a supplementary data file or as a permanent DOI’d deposition. Note that your current provision under the Australian Antarctic Data Centre 1. Is not currently accessible for me to check, and 2. sounds like it may not contain the numerical values displayed in the Figures.

e) Please cite the location of the data clearly in all relevant main and supplementary Figure legends, e.g. “The data underlying this Figure can be found in S1 Data” or “The data underlying this Figure can be found in https://doi.org/XXXX”

We expect to receive your revised manuscript within two weeks. 

*Published Peer Review History*

*Press*

Sincerely,

Roli

Roland Roberts, PhD

Senior Editor,

rroberts@plos.org,

PLOS Biology

DATA POLICY:

Regardless of the method selected, please ensure that you provide the individual numerical values that underlie the summary data displayed in the following figure panels as they are essential for readers to assess your analysis and to reproduce it: Figs 1AB, 2AB ,3AB, 4, S1AB, S2, S3ABC, S4ABC, S5ABC, S6. NOTE: the numerical data provided should include all replicates AND the way in which the plotted mean and errors were derived (it should not present only the mean/average values).

DATA NOT SHOWN?

---

## [Editor Report · Decision Letter 4]

16 Nov 2022

Dear Jasmine,

Thank you for the submission of your revised Research Article "Threat management priorities for conserving Antarctic biodiversity" for publication in PLOS Biology. On behalf of my colleagues and the Academic Editor, Andrew Tanentzap, I'm pleased to say that we can in principle accept your manuscript for publication, provided you address any remaining formatting and reporting issues. These will be detailed in an email you should receive within 2-3 business days from our colleagues in the journal operations team; no action is required from you until then. Please note that we will not be able to formally accept your manuscript and schedule it for publication until you have completed any requested changes.

Sincerely, 

Roli

Senior Editor

PLOS Biology

rroberts@plos.org